# Next generation synthetic memory via intercepting recombinase function

Andrew E. Short [1,2], Dowan Kim [1,2], Prasaad T. Milner[1] & Corey J. Wilson [1] ✉

Here we present a technology to facilitate synthetic memory in a living system via repurposing Transcriptional Programming (i.e., our decision-making technology) parts, to regulate (intercept) recombinase function post-translation. We show that interception synthetic memory can facilitate programmable loss-of-function via site-specific deletion, programmable gain-of-function by way of site-specific inversion, and synthetic memory operations with nested Boolean logical operations. We can expand interception synthetic memory capacity more than 5-fold for a single recombinase, with reconfiguration specificity for multiple sites in parallel. Interception synthetic memory is ~10-times faster than previous generations of recombinase-based memory. We posit that the faster recombination speed of our next-generation memory technology is due to the post-translational regulation of recombinase function. This iteration of synthetic memory is complementary to decision-making via Transcriptional Programming – thus can be used to develop intelligent synthetic biological systems for myriad applications.

An intelligent chassis cell can be defined as a synthetic biotic system capable of decision-making and memory operations. In the said system, decision-making is composed of one or more INPUT(s) mapped to an OUTPUT, such that the system can be reset upon the removal of the INPUT(s)—Fig. 1a. In contrast, a synthetic memory operation is not reset upon the removal of cognate INPUT(s)—i.e., memory operations retain changes in the OUTPUT state upon the removal of the cognate INPUT(s)—see Fig. 1b. The most elegant exemplars of decision-making have been demonstrated as Boolean logical operations[1–6]. Whereas, synthetic memory has been demonstrated in myriad ways—e.g., bistable toggle switches[7–9], CRISPR-based editing[10], and recombinase facilitated DNA rearrangements[11–17]. Here we are particularly interested in memory operations mediated via a subclass of recombinases collectively identified as large serine integrases[18]. Notably, this iteration of memory imparts permanent genetic changes and can be programmed to achieve both gain-of-function (GOF) and loss-of-function (LOF). Briefly, serine integrases are a class of enzymes that site-specifically bind and reconfigure sets of DNA elements—i.e., attachment sites *attB* and *attP*—resulting in DNA deletion (Fig. 1c) or DNA inversion (Fig. 1d), dependent on the orientation of the set of attachment sites. Serine integrases are an important technology and have been deployed for a

broad range of purposes[19–23]. We posit that serine integrase-based memory can be strategically paired with Transcriptional Programming[1–3,6] (i.e., our version of decision-making) to facilitate the development of intelligent chassis cells, see Supplementary Note 1. Briefly, Transcriptional Programming leverages a system of synthetic transcription factors and cognate DNA operators that can be paired with promoter elements to form inducible systems, Fig. 1e. Our synthetic transcription factors are unique in that two phenotypes can be engineered (repressors denoted as superscript + and anti-repressors denoted as superscript A), and all transcription factors can be networked via shared DNA-binding functions. Each synthetic transcription factor can be modularly designed from two fundamental parts: (1) a regulatory core domain (RCD), and (2) an engineered DNA-binding domain—i.e., alternate DNA recognition (ADR) motif, Fig. 1e. Each RCD can be abbreviated using a single letter via the nomenclature defined in Swint-Kruse et al.[24], e.g., LacI = I, GalR = G, RbsR = R, CelR = E, FruR = F. Each ADR is cognate to an engineered operator DNA element. The engineered DNA-binding functions are predicated on ADR position 17, 18 and 22 of the native LacI DNA-binding domain (i.e., residues Y17, Q18, and R22 abbreviated as YQR) being concurrently varied and paired with one or more putative symmetric operator DNA variant(s)

[1]Georgia Institute of Technology, School of Chemical and Biomolecular Engineering, Atlanta, GA, USA. [2]These authors contributed equally: Andrew E. Short, Dowan Kim. ✉e-mail: corey.wilson@chbe.gatech.edu

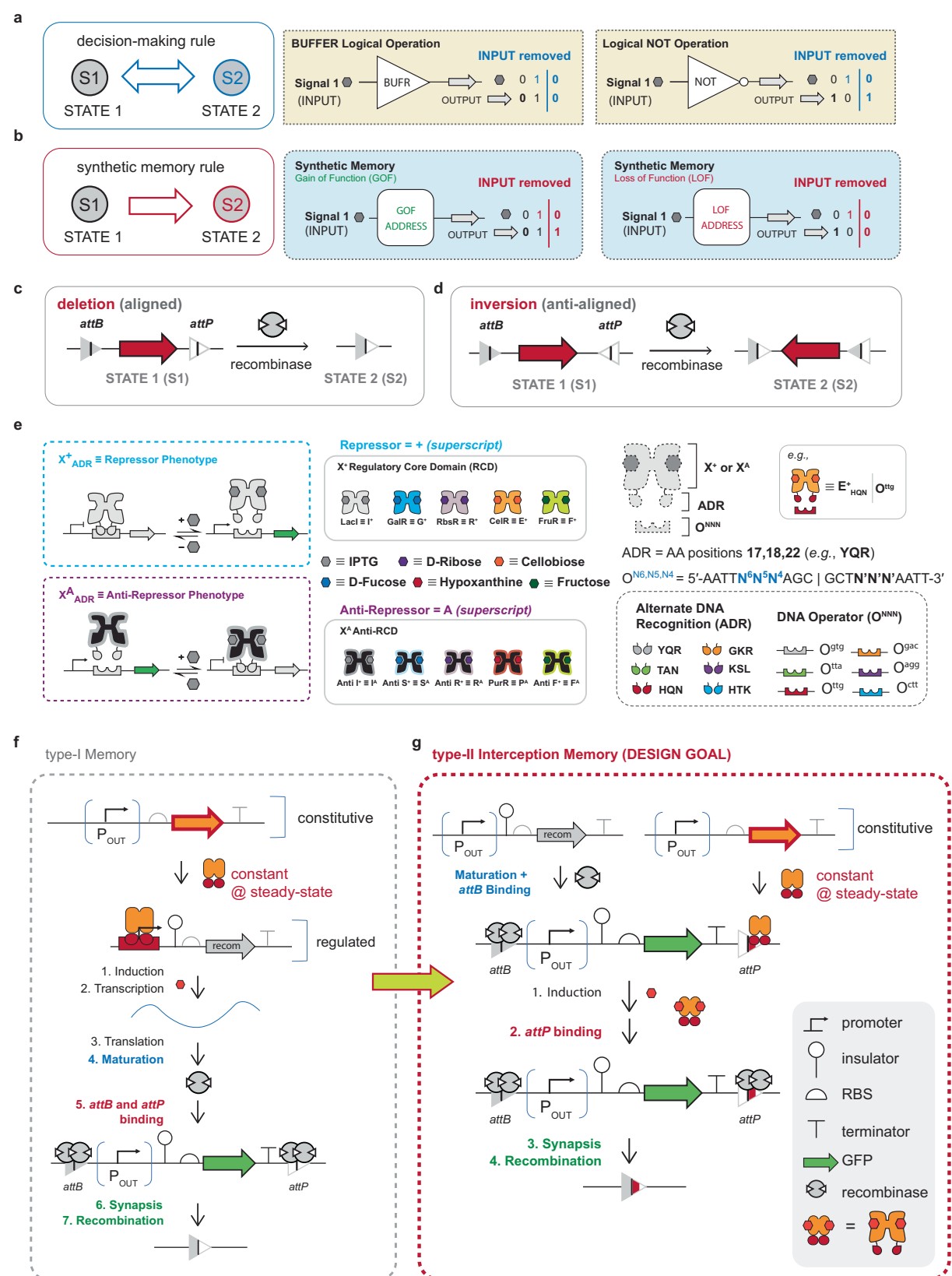

with substitutions at positions 6, 5, and 4 in the left half-site of the DNA operator. A putative DNA operator variant is defined as 5′-AATT $N^6N^5N^4$ AGC GCT $N'N'N'$ AATT-3′ where $N^\#$ is any nucleotide and $N'$ represents the nucleotide required to make the operator fully symmetric[25]. Therefore, we can abbreviate the engineered operator as $O^{N6,N5,N4}$, see Fig. 1e. Accordingly, DNA-binding domain TAN pairs with operator

DNA element $O^{tta}$—whereas HQN pairs with $O^{ttg}$ and so forth. To simplify the interpretation of the pairing of an engineered ADR with a DNA operator all cognate sets have been color-coded, see Fig. 1e.

Canonical synthetic memory (type-I) is achieved by way of the regulation of a given recombinase, which is typically induced by a small molecule. Once matured the recombinase attaches to DNA elements

**Fig. 1 | Schematic representation of interception systems. a** The decision-making rule is shown as bidirectional between STATE 1 and STATE 2. The representative logical operations in decision-making are BUFFER and NOT logical operations. When the INPUT is absent, the OUTPUT is OFF in BUFFER and ON in NOT operation (STATE 1). When the INPUT is present, the OUTPUT is ON in BUFFER and OFF in NOT operation (STATE 2). Once the INPUT is removed, the OUTPUT state reverts to STATE 1. **b** The synthetic memory rule is shown as unidirectional between STATE 1 and STATE 2. The representative synthetic memory operations are Gain of Function (GOF) and Loss of Function (LOF). The INPUT and OUTPUT state in synthetic memories are the same as decision-making, however, when the INPUT is removed, the OUTPUT state does not revert back to STATE 1. **c, d** Two types of recombination events are shown. When *attB* and *attP* are aligned, recombination results in deletion of the DNA element between *attB* and *attP* (**c**). When *attB* and *attP* are anti-aligned, recombination results in inversion of the DNA element between *attB* and *attP* (**d**). Note: The icon for the recombinase is given as a monomer. **e** Two types of

transcription factors used for interception are shown. The blue box shows the repressor mechanism, and the purple box shows the anti-repressor mechanism. The right panel illustrates the regulatory protein template. This system consists of three parts: a dimeric regulatory core domain (RCD or anti-RCD), alternate DNA recognition (ADR), and DNA operator ($O^{NNN}$). The RCD can be abbreviated as I, G, S, R, E, or F and the superscript + or A represents the repressor or anti-repressor phenotype, respectively. Each RCD has a cognate inducer shown as a colored hexagon. The ADRs are named via the mutation of amino acid positions 17, 18, and 22. DNA operators are named via nucleotide substitutions at positions 6, 5, and 4 relative to the left half-site of the operator (abbreviated as $O^{N6, N5, N4}$). Each ADR binds a cognate DNA operator shown color-coded in the bottom-right box. **f** A schematic showing the seven steps that must be completed following a type-I memory circuit's induction for recombination to occur. **g** A schematic showing the four steps that must be completed following a type-II memory circuit's induction for recombination to occur.

*attB* and *attP* and results in reconfiguration of DNA, see Fig. 1f. Here we report a post-translational strategy for controlling recombinase function, termed interception (type-II), that expands the utility of recombinases for synthetic memory operations. We define interception as the controlled blocking of any protein-DNA interaction (other than RNA polymerase) via a transcription factor (TF) that interacts with a cognate DNA operator pair in situ. In recombinase systems, we posit that interception can be achieved via strategically replacing a small segment of a recombinase attachment site with a DNA operator—see Fig. 1g. We posit that mechanistically this would result in the TF—when bound to operator DNA—sterically hindering a given recombinase from binding to a cognate attachment site. Under conditions in which the TF becomes unbound (i.e., induced) the said recombinase can attach to the DNA element and catalyze the reconfiguration of cognate DNA elements (e.g., resulting in deletion or inversion—illustration and iconography given in Fig. 1c, d and Supplementary Fig. 1). Accordingly, this iteration of synthetic memory requires two parts: (1) an operation that regulates recombinase attachment post-translation, and (2) a genetic address to define the memory function, i.e., the orientation and positioning of recombinase attachment sites *attB* and *attP*. In this study, we designed, built, and tested several iterations of interception synthetic memory using engineered repressors and anti-repressors, paired with eight orthogonal recombinases and cognate attachment sites. In this report we have engineered interception synthetic memory facilitating programmed: (1) LOF via post-translationally induced deletion, (2) GOF by way of post-translationally regulated inversion, and (3) synthetic interception memory with nested Boolean logical operations. In addition, we demonstrated that interception synthetic memory capacity can be expanded via the re-design of the central conserved region of a given set of attachment sites—allowing multiple orthogonal interception synthetic memory events via a single recombinase. Finally, we illustrated that interception regulated synthetic memory is faster than previous iterations of recombinase-based memory[17,26–28]. We posit that interception synthetic memory will enable the development of next-generation synthetic biology circuits for myriad applications in biological security, living therapeutics, biomanufacturing and the like.

## Results

### Engineering a deletion circuit with post-translational control
Recombinase attachment sites are sets of DNA elements (*attB* and *attP*), that can be partitioned—relative to a central conserved region —into four half-sites ~25–35 bp each. When half-site DNA elements are combined this results in full attachment sites on the order of ~50–70 bp per *attB* or *attP*, forming a pseudo-palindrome relative to the central motif, see Supplementary Figs. 1–3. Structural information implies that the minimum unit for a given recombinase-DNA complex is 2 half-sites and 2 recombinases[29]. In addition, studies have demonstrated that in many cases each half-site is tolerant to

mutation[30–32]. We posited that a given half-site could be omitted or modified and retain specific recombinase activity. To test this assertion, we constructed a deletion circuit in which the *att* sites corresponding to recombinase A118 are placed in the same orientation (i.e., aligned)—see Fig. 1c and Supplementary Fig. 1a—flanking a reading-frame encoding green fluorescent protein (GFP) and a constitutive promoter, contained within an *Escherichia coli* (*E. coli*) chassis cell, see Fig. 2a. Next, we omitted each of the half-sites (B1, B2, P1, P2) and exposed each circuit to the cognate A118 recombinase (see Fig. 2b and Supplementary Fig. 2a). All four half-sites were amenable to omission as each variant resulted in reduced GFP fluorescence. Next, we built and tested 7 additional omission circuits with attachment sites corresponding to recombinases TP901, Int2, Int3, Int12, Bxb1, Int5, and Int8 (see Fig. 2c–i). In this experiment we were interested in evaluating recombinase functions under moderate conditions—opposed to designing optimized circuits—accordingly, the promoter strength, RBS strength, and plasmid copy numbers were fixed (see Supplementary Note 2). In all cases half-site omission was tolerated at 1 or more position(s) in each of the aforementioned systems—except in the case of Bxb1. Ghosh et al. demonstrated that the recombinase Bxb1 cannot recombine a partial *attB* site with an *attP* site based on a gel shift assay[29], accordingly we can regard Bxb1 as a negative control.

In general, our supposition with regard to half-site omission was correct. Accordingly, we posited that a given half-site could also support a substitution with a ~16 bp DNA operator, see Fig. 1c—cognate to a given transcription factor that we engineered in a previous study[2,3]. We supposed that the modified attachment site could facilitate post-translational regulation of recombinase function, see Fig. 2j, k. Given that the A118 system displayed a high tolerance to half-site omission (Fig. 2b), we conducted a coarse-grained scan of operator substitution across the *attP* site, without modification to the central conserved dinucleotide AA (see Fig. 2j–l). The justification for initially focusing on the *attP* attachment site for substitution with operator DNA—configured as a deletion (aligned) memory circuit—was predicated on this iteration of the memory circuit facilitating the evaluation of recombinase interception as a simple function (see Supplementary Note 3). In the presented scanning experiment, the $O^{ttg}$ operator was placed at 7 disparate positions (P-24, P-18, P-15, P-14, P + 1, P + 3, and P + 4—i.e., without changing the central dinucleotide), and paired with the constitutive expression of the $E^+_{HQN}$ transcription factor and A118 recombinase. Only 3 out of 7 positions supported interception of recombinase function, and maintenance of TF induction, under the conditions tested. Namely, operator positions P-18, P-15, and P + 1 disrupted recombination in the presence of the $E^+_{HQN}$ transcription factor without inducer, and phenotypes were confirmed by complementary qualitative genotype experiments (see Supplementary Fig. 4). Upon the addition of the cellobiose ligand the TF was induced, and the attachment site was deprotected and allowed the recombinase

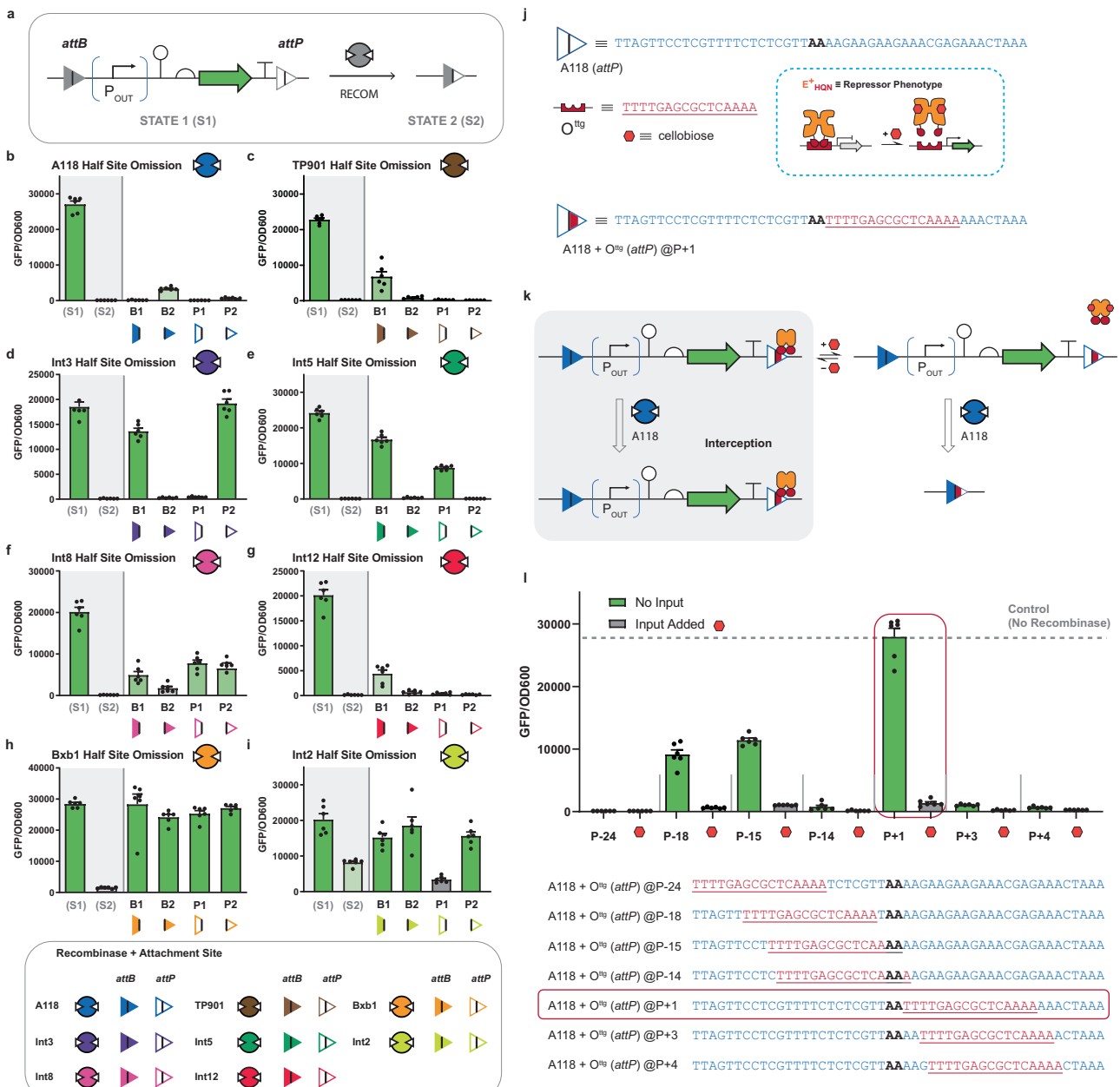

**Fig. 2 | Recombinase attachment half-site omission and design heuristics for engineering an interception synthetic memory circuit. a** Schematic of the recombinase deletion circuit, in which a reporter circuit comprised of a constitutive promoter, ribozyme, RBS, and green fluorescent protein (GFP) reading frame flanked by an aligned *attB* and *attP* pair (STATE 1). Note: We used a genetic insulator (ribozyme) to catalyze the removal of the 5′ UTR of the transcript to normalize GFP expression. A recombinase (RECOM) matched to the given att sites catalyzes recombination between *attB* and *attP*, resulting in deletion of the entire circuit (STATE 2). **b**–**i** Half-site omission tests with various recombinases are shown. For each plot, the two data bars shown in the shaded area represent controls; S1 measures fluorescence of cells transformed with the reporter plasmid alone, and S2 measures fluorescence of cells transformed with the reporter plasmid plus the corresponding constitutive recombinase expression plasmid. In the unshaded areas half-site omissions are shown with the cognate recombinases present. B1 refers to a construct in which the first half of the *attB* site has been omitted, B2

refers to a construct where the second half of the *attB* site has been omitted, likewise half-site omissions P1 and P2 correspond to positions in the *attP* site. Details for each recombinase are given in Supplementary Figs. 2 and 3. **j** A granular description of an attachment site substituted with a 16 base pair operator. In this example the $O^{ttg}$ operator is substituted within an *attP* site that corresponds to recombinase A118. The $O^{ttg}$ DNA operator is cognate to the $E^+_{HQN}$ transcription factor. **k** Schematic of the putative mechanism of interception; in the gray box at left, repressor binding at the $O^{ttg}$ DNA operator (within the *attP* site) protects the circuit from deletion catalyzed by the A118 recombinase. To the right, when the repressor is induced, it unbinds from the att site, and A118 can recombine and delete the circuit. **l** A variety of operator positions were tested, identifying the P + 1 location as the best candidate for controlling recombination with interception. Source data are provided as a Source Data file. Data in (**b**–**i**) and (**l**) represent the average of *n* = 6 biological replicates. Error bars correspond to the SEM of these measurements.

to delete the GFP circuit. The putative mechanism for interception is given in Fig. 2k. The P + 1 iteration of this circuit exhibited the best performance. These results allowed us to glean our first design rule, inferring that the position of an operator can impact interception performance.

## Engineering next-generation memory circuits using transcription factors with alternate DNA binding

Given the high performance of the P + 1 iteration of the interception memory circuit—composed of the A118 *attP* attachment site substituted with the $O^{ttg}$ operator and of $E^+_{HQN}$ regulator—we posited that

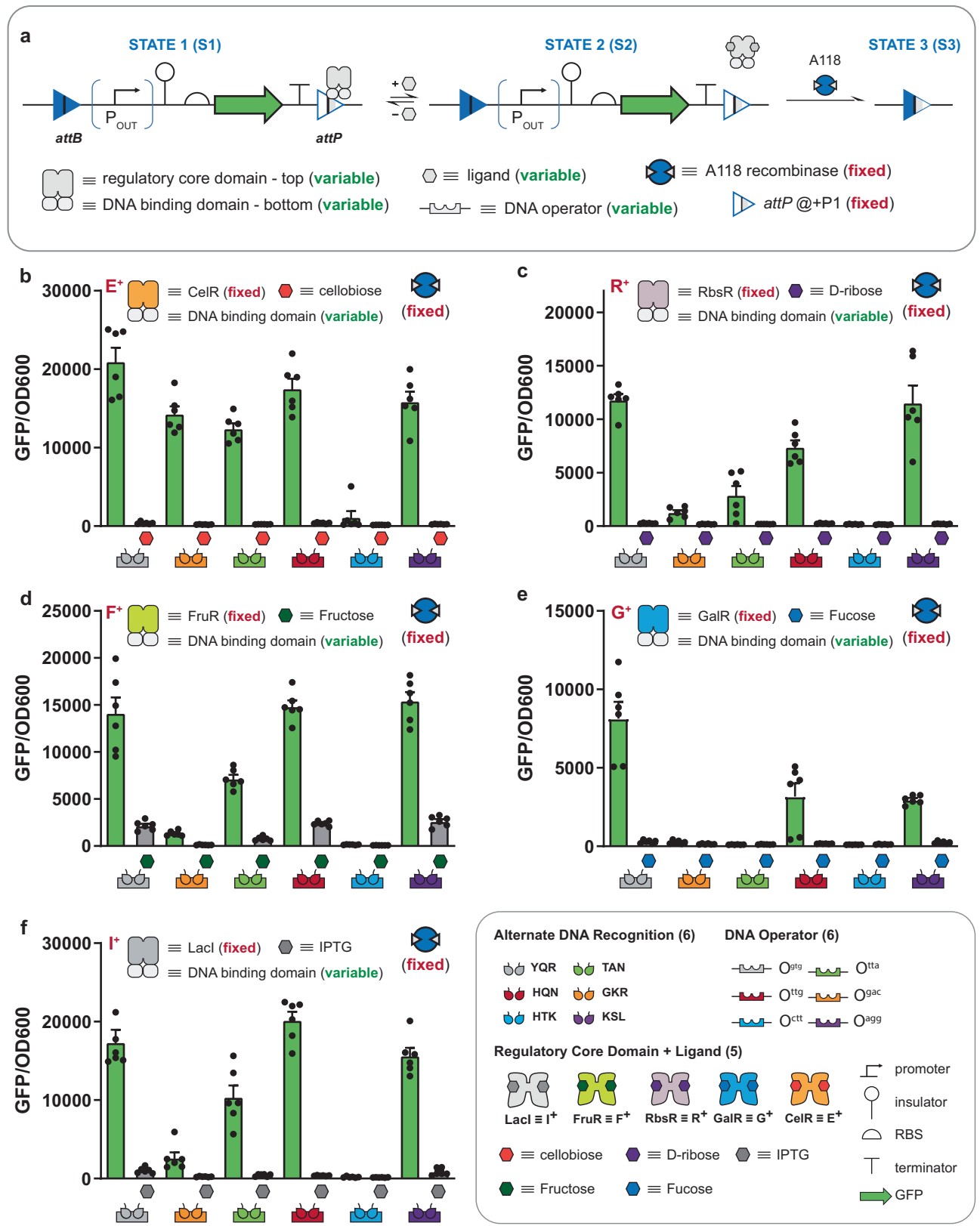

we could systematically vary the DNA operator element at this fixed position (see Fig. 3a). Here we maintained the CelR (E⁺) regulatory core domain, while varying the DNA-binding function using 5 additional alternate and orthogonal DNA operator elements (i.e., $O^{gtg}$, $O^{gac}$, $O^{tta}$, $O^{ctt}$, $O^{agg}$), see Fig. 3b. In this iteration of the experiment the promoter strength, RBS strength, and plasmid copy numbers were fixed (see

Supplementary Note 2). In all cases (apart from the $E^+_{HTK} | O^{ctt}$ set) the said interception memory circuits with alternate TF binding were functional. Specifically, the substitution of $O^{ttg}$ with any of the following operators $O^{gtg}$, $O^{gac}$, $O^{tta}$, or $O^{agg}$ and cognate TFs (i.e., $E^+_{YQR}$, $E^+_{GKR}$, $E^+_{TAN}$, or $E^+_{KSL}$, respectively) disrupted recombination in the absence of cellobiose—observed as an intact reading frame with GFP expressed.

**Fig. 3 | Interception synthetic memory with expanded information processing and operator variation. a** A schematic summarizing the mechanism and genetic construct (deletion circuit) used to assess A118 recombinase interception with variable repressors directed at different operators placed in the P + 1 position (described in Fig. 2j). The A118 recombinase and relevant repressor are constitutively expressed in all cases. The repressors used are comprised of two modular domains: (1) a regulatory core domain that allows the repressor to be induced by a different ligand, and (2) a DNA-binding domain that allows the repressor to bind to a different operator. In STATE 1, repressor binding at the operator blocks recombinase function, protecting the circuit from deletion. Inducing the repressor brings the circuit to STATE 2, where the recombinase can access the *attP* site to recombine the circuit (bringing the circuit to STATE 3). **b** Assay data for intercepted (minus ligand) circuits vs. deprotected (induced) circuits using the repressor E$^+$ across six different DNA-binding domain/operator pairs. Assay data using the same set of DNA-binding domain/operator pairs as (**b**) paired with different regulatory core domains as follows: **c** R$^+$, **d** F$^+$, **e** G$^+$, and **f** I$^+$. (inset) Symbols for the circuit parts and modular repressor components used, including six DNA-binding domains conferring alternate DNA recognition, the six DNA operators where those DNA-binding domains can bind (color-matched), and the five regulatory core domains sensitive to different ligands. Source data are provided as a Source Data file. Data in (**b**–**f**) represent the average of *n* = 6 biological replicates. Error bars correspond to the SEM of these measurements.

However, upon the addition of cellobiose each E$^+_{ADR}$ transcription factor—with alternate DNA recognition (ADR)—was induced and the deletion of the GFP circuit was observed. Initially, we posited that the E$^+_{HTK}$ | O$^{ctt}$ at P + 1 memory circuit failed due to low DNA-binding affinity as both states with and without the ligand were deleted. Assessment of the general binding of each of the TF supports this initial supposition from the vantage point of dynamic range and the extent of leakiness in the bound state—with few exceptions (see Supplementary Figs. 5 and 6). While the general performance of the E$^+_{HTK}$ transcription factor was the lowest in terms of dynamic range; however, the leakiness was *on par* with the E$^+_{TAN}$ | O$^{tta}$ transcription factor. Accordingly, we revised our supposition and posited that in addition to the transcription factor performance the composition of the attachment site may also impact the performance of an interception circuit. A sequence alignment of the substituted *attP* sites relative to the wild-type attachment site revealed that the *attP* with the O$^{ctt}$ substitution and the *attP* with the O$^{gac}$ substitution had the highest sequence similarity to wild-type *attP* (see Supplementary Fig. 5). This observation implies that the *attP* O$^{ctt}$ substituted site—perhaps followed by the *attP* O$^{gac}$ substituted site—is likely catalytically more efficient with respect to recombination relative to the other modified *attP* sites, rationalizing the observation for the performance of the E$^+_{HTK}$ | O$^{ctt}$ interception circuit. Accordingly, we articulated our second design rule, which purports that transcription factor DNA-binding affinity impacts interception efficiency—and this property can be confounded by the variation of the sequence of the *attP* site, which could impact recombination catalytic efficiency. In addition to evaluating the general performance of regulated synthetic memory circuits, we evaluated interception over 3 days. We evaluated 3 systems regulated by (1) E$^+_{HQN}$ | O$^{ttg}$, (2) E$^+_{YQR}$ | O$^{gtg}$, and (3) E$^+_{KSL}$ | O$^{agg}$ as exemplars of the longitudinal stability of intercepted recombinase function. In all cases, as time increased the stability of the protected circuit was evidenced, see Supplementary Table 1.

### Engineering next-generation memory circuits with expanded INPUT processing capability

We designed built and tested memory circuits with different engineered transcription factors to facilitate expanded INPUT processing (see Fig. 3c–f). Initially, we fixed the operator position (i.e., P + 1 with respect to the A118 *attP* attachment site) and fixed the composition of the operator substitution (i.e., O$^{gtg}$ DNA element—cognate to the YQR binding domain), while varying the regulatory core domain of the TF to enable processing of 4 additional inputs, i.e., I$^+_{YQR}$ (IPTG), R$^+_{YQR}$ (ribose), F$^+_{YQR}$ (fructose), G$^+_{YQR}$ (fucose). Qualitatively, all four memory circuits performed as expected such that un-induced systems reduced circuit deletion, while induced TFs with cognate ligands de-protected the *attP* site and resulted in circuit deletion. We posited that the quantitative differences in deletion between circuits could be attributed to differences in TF-operator affinity—congruent with the second design rule (see Supplementary Figs. 7–11).

In the next iteration of the memory circuits with expanded INPUT processing, we varied the DNA operator for each of the given TFs (i.e., O$^{gac}$, O$^{tta}$, O$^{ttg}$, O$^{ctt}$, and O$^{agg}$—cognate to ADR domains GKR, TAN, HQN, HTK, and KSL respectively) and evaluated interception. In summary, putative interception memory circuits now contain two variables: (1) the TF and (2) the DNA operator element (see Fig. 3c–f). Congruent with the A118 attachment sites substituted at position P + 1 with alternate operators and cognate E$^+_{ADR}$ regulators (Fig. 3b), the majority of said interception memory circuits with alternate TF and operator binding were functional—except for HTK | O$^{ctt}$ (in all cases). Moreover, the *attP* O$^{gac}$ substituted site was the second least effective memory circuit even when the general binding metrics would imply a reasonable probability for interception (e.g., I$^+_{GKR}$ | O$^{gac}$, and R$^+_{GKR}$ | O$^{gac}$, see Supplementary Figs. 7 and 10). This observation is consistent with our expectation given the purported differences in catalytic efficiency for the *attP* O$^{ctt}$ substituted and *attP* O$^{gac}$ substituted attachment sites based on the sequence alignment—even after accounting for the poor binding performance of TFs from the HTK and GKR sets (also see Supplementary Figs. 7–11). To affirm interception memory at the population level we used flow cytometry for a subset of best performing operations, see Supplementary Fig. 12. In general, we demonstrated that interception was capable of protecting the attachment site—completely in many cases, and deprotected memory circuits resulted in recombination for the entire population of unrecombined cells, in nearly every case. All tested interception memory circuits performed according to the supposition posited by our second design rule. Namely, the (1) performance metrics for each TF and (2) qualitative catalytic efficiency of modified attachment sites dictate the performance of a given synthetic (interception) memory circuit. However, (3) increased catalytic efficiency of a given *attP* site can be reduced with sufficient binding function of the TF (see Supplementary Note 4).

To demonstrate that interception memory circuits can be tuned to improve performance we selected three variants from the RbsR set with the poorest performance, i.e., (1) R$^+_{GKR}$ | O$^{gac}$, (2) R$^+_{TAN}$ | O$^{tta}$, and (3) R$^+_{HTK}$ | O$^{ctt}$. Noting that all three operator substituted *attP* sites had the highest sequence similarity to the wild-type attachment site (see Supplementary Fig. 5) we posited that we could improve the circuit performance via diminishing the apparent recombinase (A118) activity. To accomplish this, we reduced the RBS strength cognate to A118 production effectively reducing the amount of catalyst available for recombination, see Supplementary Fig. 13. In all cases, we observed a marked improvement in circuit performance for all three iterations of synthetic memory circuits—affirming our supposition.

### Engineering permissive interception memory circuits—INPUT processing via anti-repression

We recently engineered a collection of anti-repressors—i.e., anti-LacI (I$^A$)[2,33], anti-RbsR (R$^A$)[2], anti-FurR (F$^A$)[2], anti-GalS (S$^A$)[34], and PurR (P$^A$)[35]—that are phenotypically antithetical (see Fig. 1e) to many of the repressors used in Fig. 3. In the context of gene regulation, repressors function as BUFFER operations, whereas anti-repressors function as NOT operations—descriptions given in Fig. 1a, e. Accordingly, we can describe the anti-repressor phenotype as permissive in that DNA binding is only permitted in the presence of the INPUT ligand. We posited that we could engineer permissive interception memory

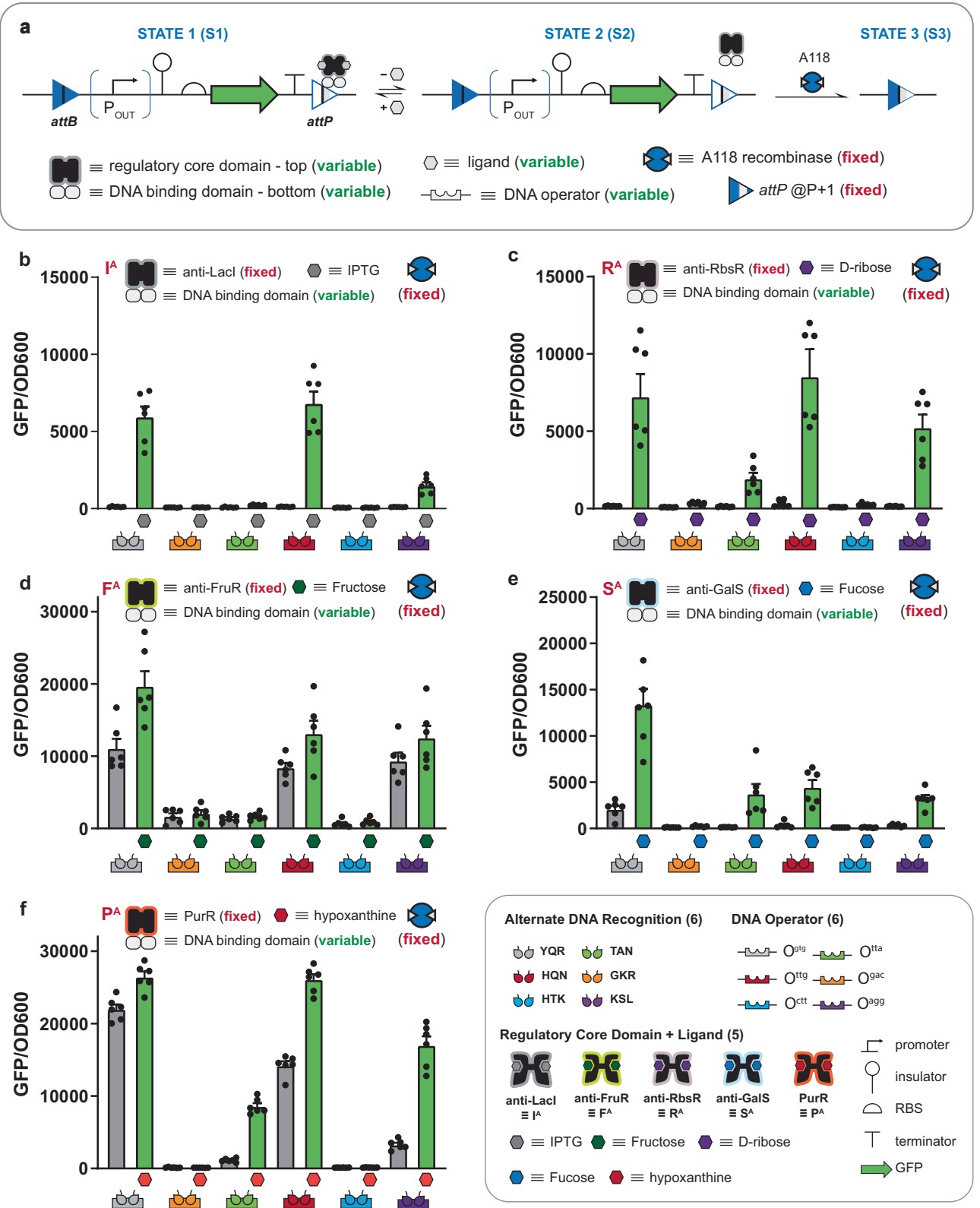

operations that could retain the reading frame of a given circuit only if the cognate exogenous signal was present—general description given in Fig. 4a. Initially, we constructed deletion circuits with the $O^{ttg}$ operator substituted within the A118 *attP* half-site at the P + 1 position and paired the circuit with one of the given anti-repressors, i.e., $I^A_{HQN}$ | $O^{ttg}$ (Fig. 4b), $R^A_{HQN}$ | $O^{ttg}$ (Fig. 4c), $F^A_{HQN}$ | $O^{ttg}$ (Fig. 4d), $S^A_{HQN}$ | $O^{ttg}$ (Fig. 4e), and $P^A_{HQN}$ | $O^{ttg}$ (Fig. 4f). In the absence of ligand, the tested anti-repressors allowed the cognate recombinase to delete the circuit

(see Fig. 4 and Supplementary Figs. 14–18). However, in the presence of the cognate ligands, i.e., IPTG ($I^A_{HQN}$ | $O^{ttg}$), ribose ($R^A_{HQN}$ | $O^{ttg}$), fructose ($F^A_{HQN}$ | $O^{ttg}$), fucose ($S^A_{HQN}$ | $O^{ttg}$), and hypoxanthine ($P^A_{HQN}$ | $O^{ttg}$)—memory circuits exhibited induced protection (interception) observed as maintenance of the GFP circuit. To demonstrate generalizability, we replaced the $O^{ttg}$ operator with five additional operators ($O^{gac}$, $O^{tta}$, $O^{gtg}$, $O^{ctt}$, and $O^{agg}$) corresponding to binding motifs GKR, TAN, YQR, HTK, and KSL respectively, i.e., the same set used for the repressors tested in

**Fig. 4 | Permissive interception synthetic memory via anti-repression. a** A schematic summarizing the mechanism and genetic construct (deletion circuit) used to assess A118 recombinase interception for different anti-repressors directed at different operators placed in the P + 1 position (described in Fig. 2j). Anti-repressors have the opposite induction/DNA-binding relationship to repressors; anti-repressors bind DNA when induced and do not bind DNA when not induced. The A118 recombinase and relevant anti-repressor are constitutively expressed in all cases. As in Fig. 3, the anti-repressor used are comprised of two modular domains: (1) a regulatory core domain that allows the anti-repressor to be induced by a different ligand, and (2) a DNA-binding domain that allows the anti-repressor to bind to a different operator. In STATE 1, induced anti-repressor binding at the operator blocks recombinase function, protecting the circuit from deletion. In the absence of ligand, the anti-repressor cannot bind to the operator, bringing the circuit to STATE 2, where the recombinase can access the *attP* site to recombine the circuit (bringing the circuit to STATE 3). **b** Assay data for deprotected (minus ligand) circuits vs. intercepted (induced) circuits using I$^{A(5)}$ across six different DNA-binding domain/operator pairs. Assay data using the same set of DNA-binding domain/operator pairs as (**b**) paired with different regulatory core domains as follows: **c** R$^{A(1)}$, **d** F$^{A(1)}$, **e** S$^{A(1)}$, and **f** P$^A$. (inset) Symbols for the circuit parts and modular anti-repressor components used, including six DNA-binding domains conferring alternate DNA recognition, the six DNA operators where those DNA-binding domains can bind (color-matched), and five regulatory core domains sensitive to different ligands. Source data are provided as a Source Data file. Data in (**b–f**) represent the average of $n = 6$ biological replicates. Error bars correspond to the SEM of these measurements.

Fig. 3. In addition to synthetic transcription factors X$^A_{HQN}$ (where X = I, R, F, S or P) X$^A_{YQR}$ and X$^A_{KSL}$ facilitated some degree of permissive protection of the deletion circuit – i.e., in the presents of the given cognate ligands. Moreover, similar to the engineered repressors adapted with the HTK binding motif (i.e., X$^+_{HTK}$ where X = I, R, F, G or E, see Fig. 3) used for type II memory, all engineered X$^A_{HTK}$ anti-repressors with the said DNA-binding function failed to protect the deletion circuit, see Fig. 4. However, in addition to X$^+_{HTK}$, anti-repressors adapted with GKR (X$^A_{GKR}$) also failed to permissively protect the deletion circuit. Based on these observations, we concluded that the permissive maintenance of a deletion circuit via interception is possible. However, given the mechanism of protection (i.e., expression and folding of the TF precedes ligand binding, which is required for interaction with the substituted operator at the attachment site) fewer successful operations were observed. Notably, decreasing the RBS strength to produce less recombinase did not necessarily improve interception synthetic memory (see Supplementary Fig. 19). Rather, tuning the RBS resulted in overprotected circuits, at best. This observation implies that improving the performance of permissive synthetic memory circuits would likely require additional protein engineering or pre-conditioning, opposed to circuit optimization. Finally, we can bring together inducible and permissive memory with nested logical operations to form systems capable of decision-making and memory operations, see Supplementary Figs. 20–22.

### Engineering next-generation memory circuits with orthogonal recombinase functions

To this point, we have only focused on the development of interception memory circuits using the A118 recombinase and cognate attachment sites. Noting the general performances of half-site omissions for other recombinases (Fig. 2) and the A118 exemplars given in Figs. 3 and 4, we posited that we could successfully construct P + 1 iterations of interception memory circuits responsive to additional recombinase functions—in the context of a deletion memory circuit (general design given in Fig. 5a). First, we designed, built and tested 7 new circuits with attachment sites that corresponded to recombinases TP901, Int2, Int3, Int12, Bxb1, Int5 and Int8 (see Fig. 5c–i—gray boxes). For this set of circuit designs, we substituted the DNA operator O$^{ttg}$ (cognate to E$^+_{HQN}$) at the P + 1 position within the *attP* site for each of the 7 additional recombinases (illustrated in Fig. 5a). The rationale for selecting the O$^{ttg}$ operator (cognate to the E$^+_{HQN}$ transcription factor) and the P + 1 position was that this iteration of the interception memory circuit had one of the highest performances when tested with the A118 recombinase (see Figs. 2, 3 and 5b). Next, we evaluated each of the P + 1 deletion circuits with and without the corresponding recombinase (i.e., unregulated), see Fig. 5—gray boxes. Briefly, 5 out of the 7 additional circuits resulted in deletion upon the concurrent production of the cognate recombinase, i.e., excluding Bxb1 and Int2. As expected, the Bxb1 P + 1 circuit showed abrogated function upon modification, congruent with results shown in Fig. 2h. In the case of the Int2 deletion circuit, we posited that given the performance of the P2

half-site omission (see Fig. 2i) the operator substitution at the P + 1 position compromised recombinase function.

Next, we constructed the corresponding regulated memory circuits with the E$^+_{HQN}$ transcription factor present, and concurrent expression of a given recombinase (cognate to the attachment site). Briefly, the regulated memory circuits with demonstrated recombinase function displayed the correct qualitative performances (see Fig. 5—red outlined boxes). Namely, the transcription factor E$^+_{HQN}$ intercepted recombinase mediated deletion in all 5 functional circuits, i.e., TP901, Int3, Int12, Int5 and Int8. Upon induction with cellobiose a given circuit was deprotected and deletion ensued for said functional circuits. Finally, we tested variation in the position of the DNA operator within the *attP* site (see Supplementary Fig. 23). In general, interception was observed in at least one additional site for at least 5 of the functional systems, based on a coarse-grained scan. However, none of the previously non-functional systems (i.e., Bxb1 and Int2) exhibited recovery. From this set of experiments, we gleaned that: (1) in general variation of the attachment site is tolerated, (2) however, TF binding can confound interception outcomes, (3) and multiple operator positions can be supported for a given *attP* site and correlated with tolerance to general half-site omission.

### Interception with combinational (2-INPUT) information processing

We originally developed our system of synthetic transcription factors to work in collaboration, forming 2-INPUT gene control (decision-making) from fundamental single input operations—facilitated via directing two or more non-synonymous transcription factors to the same DNA element[1–3]. Likewise, we posited that we could build 2-INPUT interception memory circuits, using similar design principles (see Fig. 6a). In our first iteration of a 2-INPUT memory circuit we paired the E$^+_{YQR}$ and I$^+_{YQR}$ repressors via the O$^{gtg}$ operator element (see Fig. 6a, b). Congruent with our design goal the circuit remained intact without ligand or only with one ligand present. Deletion of the circuit ensued only when the system was exposed to both signals concurrently. To demonstrate the generalizability of 2-INPUT interception memory we constructed four additional iterations with variation and ligand response and DNA-binding function, see Fig. 6c–f. Next, we constructed an antithetical interception memory operation using two anti-repressors that processed disparate inputs, with synonymous DNA-binding functions, see Fig. 6g. In the said memory system deletion was only possible in the absence of both input signals. Finally, we constructed a mixed interception operation in which we paired a repressor with an anti-repressor, see Fig. 6h. In this iteration, deletion of the circuit was only possible in the presence of cellobiose and was intercepted in all other cases.

### Interception synthetic memory with inversion addresses

We posited that we could design an iteration of interception in which the attachment site configuration would facilitate inversion (opposed to deletion) upon induction of a cognate repressor. This iteration of

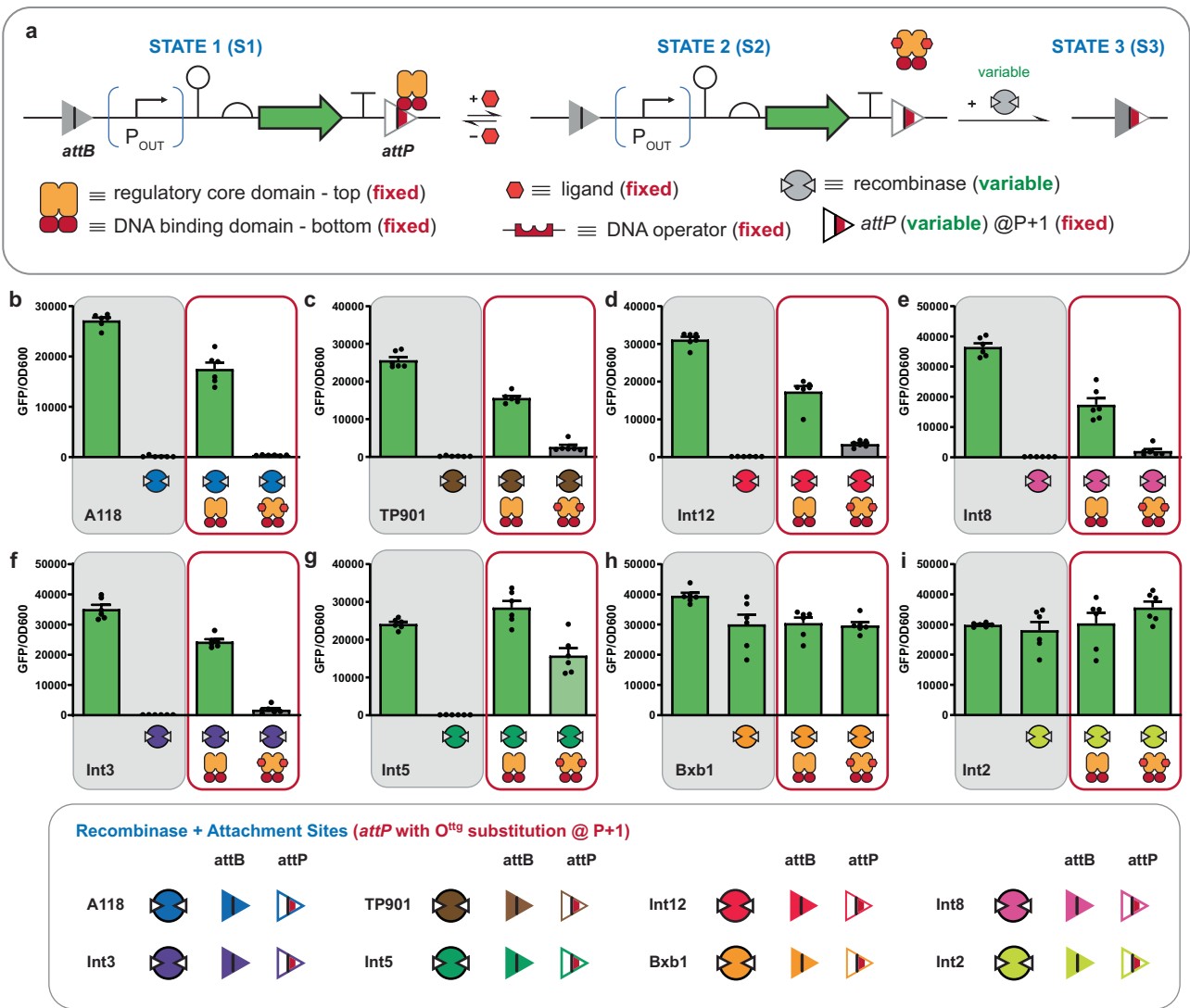

**Fig. 5 | Interception via orthogonal recombinase functions. a** A schematic summarizing the mechanism and genetic construct (deletion circuit) used to assess interception for eight different recombinases regulated by $E^+_{HQN}$ directed to the cognate operator substituted at the P + 1 position for each recombinase (also see Fig. 2 and Supplementary Figs. 2 and 3). The relevant recombinase and $E^+_{HQN}$ are constitutively expressed in all cases. As in Fig. 3, in STATE 1, $E^+_{HQN}$ binding at the operator blocks recombinase function, protecting the circuit from deletion. Inducing the repressor brings the circuit to STATE 2, where the recombinase can access the *attP* site to recombine the circuit (bringing the circuit to STATE 3). **b** Data in the gray box represent control data; the left bar displays data for *E. coli* cells containing the reporter (GFP) plasmid alone, representing maximum fluorescence. The right bar displays data for *E. coli* cells containing the reporter and recombinase expression plasmids. However, the $E^+_{HQN}$ repressor is not present accordingly interception is not possible, thus the circuit is deprotected. Data in the red box display the effect of interception on the circuit for *E. coli* containing all three plasmids (reporter, recombinase, and repressor); on the left is the intercepted (minus ligand) circuit, and on the right is the deprotected (induced) circuit. Assay data following the same format as (**b**) for different recombinases as follows: **c** TP901, **d** Int12, **e** Int8, **f** Int3, **g** Int5, **h** Bxb1, and **i** Int2, also see Supplementary Fig. 4 for qualitative genotype data. Inset at bottom is a schematic defining the colors for different recombinases and their cognate attachment sites. Source data are provided as a Source Data file. Data in (**b**–**i**) represent the average of *n* = 6 biological replicates. Error bars correspond to the SEM of these measurements.

interception memory required the anti-alignment of attachment sites, see general designs given in Supplementary Fig. 1b. The envisioned inversion memory circuit would facilitate regulated and inheritable gain-of-function, in contrast to the regulated loss-of-function facilitated by demonstrated deletion interception circuits. Here the design required that we take into consideration the placement of the DNA operator to prevent canonical gene regulation. In our first iteration we substituted the operator within the *attP* site (cognate to the A118 recombinase) at the P + 1 position, such that upon inversion the operator is distal (far upstream) to the promoter (see Fig. 7a). Noting that final placement of the operator proximal to the promoter would likely result in regulation of RNA polymerase readthrough. Congruent with the circuit design, A118 recombinase activity was intercepted in the absence of ligand (i.e., the promoter remained inverted in the presence

of recombinase). Upon the induction of the $I^+_{HQN}$ transcription factor the attachment site became deprotected and the promoter was inverted facilitating the production of GFP. To demonstrate the generalizability of the design we built and tested three additional iterations of inversion interception circuits for different recombinases, i.e., Int3 (Fig. 7b), Int8 (Fig. 7c), and Int12 (Fig. 7d), with variation in operator placement or attachment site configuration. All tested inversion interception circuits performed as expected. Moreover, upon the removal of the ligand the inverted circuit maintained the ON-state verifying that the transcription factor was divorced from gene regulation.

## Expanding synthetic memory capacity
In addition to the above, we posited that we could expand interception synthetic memory via modifying the central motif of a given set of

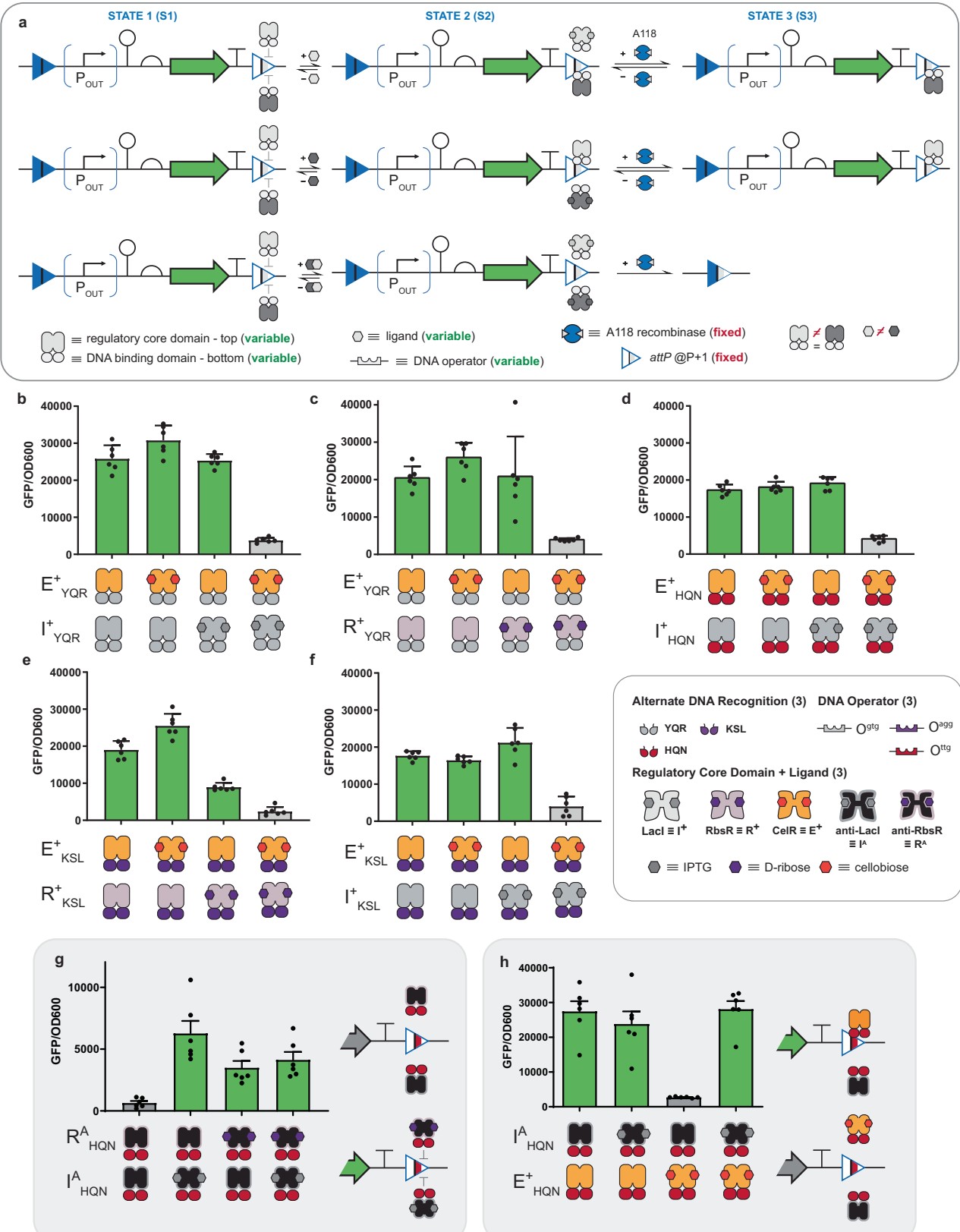

**Fig. 6 | Multiple input interception memory. a** Schematic of interception of a deletion circuit by two transcription factors (TFs). The regulatory core domains and DNA-binding domains vary, and corresponding ligands and DNA operators are used in each case. A118 recombinase is constitutively expressed in all cases, and the variable operators are always placed at the P + 1 position. The state diagram presents the relationship between repressor, ligand, operator, and recombinase with each set of ligands added. **b–f** Interception with two repressors. **b** $E^+_{YQR}$ and $I^+_{YQR}$ binding to $O^{SYM}$. **c** $E^+_{YQR}$ and $R^+_{YQR}$ binding to $O^{SYM}$. **d** $E^+_{HQN}$ and $I^+_{HQN}$ binding to $O^{ttg}$. **e** $E^+_{KSL}$ and $R^+_{KSL}$ binding to $O^{agg}$. **f** $E^+_{KSL}$ and $I^+_{KSL}$ binding to $O^{agg}$. **g** Interception with two anti-repressors, $R^A_{HQN}$ and $I^A_{HQN}$ binding to $O^{ttg}$. The inset schematic shows the relationship between anti-repressors, ligand, and operator. **h** Interception with one repressor, $E^+_{HQN}$, and one anti-repressor, $I^A_{HQN}$, binding to $O^{ttg}$. Source data are provided as a Source Data file. Data in (**b–h**) represent the average of $n = 6$ biological replicates. Error bars correspond to the SEM of these measurements.

attachment sites. This supposition is predicated on pairs of attachment sites that share the same central motif can recombine, whereas mismatched central motifs cannot recombine[17,36]. In other words, recombinases require identical central conserved sites for recombination, thus varying the central conserved region of the attachment sites purportedly generates orthogonal attachment site pairs. We posited that we could vary the central motif (to generate orthogonal attachment sites) and concurrently substitute an operator within the *attP* at position P + 1 (to facilitate interception). Combining attachment site orthogonality and non-synonymous interception—in principle—can facilitate the systematic expansion of this iteration of synthetic memory. To test this assertion we designed, built, and tested all 6 putatively orthogonal variations to the central motif of attachment site A118 pairs with an $O^{ttg}$ operator at position P + 1 (see Fig. 8a). Upon testing each of the deletion memory circuits and corresponding mismatches we affirmed our supposition that interception can be paired with variation in the central motif.

Next, we tested memory circuits with (1) variation in the central dinucleotide for A118 attachment sites paired with (2) variation in the substituted operator DNA (see Supplementary Figs. 24 and 25). Here we illustrated that paired orthogonal DNA elements—i.e., orthogonal A118 attachment sites + orthogonal operators—and disparate transcription factors could regulate interception in putatively orthogonal memory circuits. In turn, we used this analysis to identify the two best performing sets of putatively orthogonal fundamental interception memory operations (see Supplementary Fig. 24). Our testing and analysis revealed that the (1) $E^+_{HQN}$ | *attP* A118-CA and (2) $I^+_{KSL}$ | *attP* A118-AA single input circuits had outstanding performance and were putatively orthogonal in terms of input signal and attachment site recombination. We posited that using this set of fundamental memory operations we could construct a 2-OUTPUT interception memory circuit such that $E^+_{HQN}$ | *attP* A118-CA corresponded to mKate regulated deletion and $I^+_{KSL}$ | *attP* A118-AA corresponded to GFP regulated deletion (see Fig. 8b). The purpose of this memory circuit was to demonstrate that interception orthogonality was possible with a single recombinase facilitated by orthogonal sets of attachment sites. Congruent with our supposition we were able to demonstrate that each deletion occurred independently and was not confounded by unintended putative states, e.g., inversions or off-target deletions (see Fig. 8b and complementary flow cytometry data given in Supplementary Figs. 26 and 27).

### Synthetic memory kinetics
In this final experiment, we used the same set of parts to construct type-I and type-II memory circuits (i.e., identical promoters, insulators, RBS, and GFP OUTPUT) to maintain consistent gene expression and putative burden on the chassis cell. The only difference between the memory circuits is the design. Namely, for type-I memory the $E^+_{YQR}$ transcription factor regulated the expression of the recombinase, whereas the $E^+_{YQR}$ transcription factor regulated the recombination event directly (post-translation) in type-II memory, Fig. 8c, d and Supplementary Figs. 28–30. In all cases interception memory was significantly faster than the corresponding canonical (type-I) design, i.e., occurring nearly instantaneously opposed to hours or days. We attributed the increased rate of type-II memory over type-I memory to: (1) the maintenance of high levels of mature recombinase at steady-state, which (2) leads to near instantaneous binding to the *attB* site. Moreover, the TF and recombinase are in a dynamic equilibrium at the substituted *attP* site. Once the TF ($E^+_{YQR}$) is induced, (3) A118 recombinase binding and subsequent recombination occur near instantaneously.

## Discussion
Canonical (type-I) synthetic recombinase-based memory—illustrated in Fig. 1f—has been developed and leveraged for a variety of applications[19–23]. Type-I synthetic memory has been demonstrated as a versatile and stable technology for bestowing permanent and inheritable changes to DNA in myriad synthetic systems[18]. However, type-I synthetic memory can take hours to days to fully develop in a population of chassis cells. Moreover, type-I synthetic memory is restricted by the concurrent processing of pairs of attachment sites. For example, given two sets of att sites (with non-synonymous sets of central dinucleotides) that can be reconfigured via recombinase A118, both sets of sites will be reconfigured at the same time and cannot be decoupled or deconvoluted.

In this study we present an iteration (type-II) of synthetic biological memory (Fig. 1g) that is expandable and more versatile relative to canonical (type-I) synthetic recombinase-based memory. For example, given 6 orthogonal attachment sites that correspond to A118 and 5 orthogonal synthetic transcription factors, we can (in principle) expand the capacity of a single recombinase at least 5-fold. This advance over the state-of-the-art can support concurrent multiple aligned and anti-aligned attachment site orientations—noting that type-I memory can only support one function at a time. In addition, interception is significantly faster than canonical recombinase-based memory and supports multiple synthetic transcription factors—in addition to natural transcription factors, see Supplementary Fig. 31. In conclusion, this study represents the next-generation of synthetic biological memory, capable of greater programming capacity—executed via expanded circuits that utilize fewer cellular resources.

## Methods
### Strains and media
Standard DNA cloning was performed via chemical transformation in NEB DH5-α Chemically Competent *Escherichia coli* (*huA2 Δ(argF⁻ lacZ) U169 phoA glnV44 φ80Δ(lacZ) M15 gyrA96 recA1 relA1 endA1 thi-1 hsdR17*; New England Biolabs (NEB)), and assay experiments were performed via chemical transformation in *E. coli* strain 3.32 (*lacZ13(Oc) lacI22 λ⁻ el4- relA1 spoT thiE1*; Yale CGSC #5237). For transformations, cells were grown in SOC medium (Fisher Scientific); media for growing anti-repressor constructs required supplementation with ligands as described below. For precultures prior to assays, cells were recovered in Luria Broth (LB) Miller Medium (Fisher Scientific) supplemented with appropriate antibiotics; again, media for growing anti-repressor constructs was supplemented with appropriate ligands. Assays were performed in 1X M9 Minimal Medium (6.8 g l⁻¹ Na₂HPO₄, 3.0 g l⁻¹ KH₂PO₄, 0.5 g l⁻¹ NaCl, 1.0 g l⁻¹ NH₄Cl, 2 mM MgSO₄, 100 μM CaCl₂; Millipore Sigma) supplemented with 0.2% (w/v) casamino acids (VWR Life Sciences), 1 mM thiamine HCl (Alfa Aesar), and 0.4% (w/v) glucose. LB Miller Agar (Fisher Scientific) was used for selection during cloning. Antibiotics and ligands were used where appropriate. Antibiotics used were: chloramphenicol (25 μg ml⁻¹; VWR Life Sciences), kanamycin (35 μg ml⁻¹; VWR Life Sciences), and carbenicillin (100 μg ml⁻¹; Teknova). Ligands used were: adenine (as a precursor to the ligand for PurR, hypoxanthine[35], Acros Organics), cellobiose (Arcos Organics), D-fucose (Carbosynth), D-ribose (Arcos Organics), D-fructose (Arcos Organics), and isopropyl-β-D-thiogalactoside (IPTG; Millipore Sigma). Adenine (hypoxanthine) was supplemented in assay and anti-repressor growth media (including SOC for transformations and LB Miller agar selection plates) at 1 mM concentration; all other ligands were added to assay media at 10 mM concentration.

### Cloning and plasmid construction
For all cloning experiments, oligomer and genestrand synthesis and DNA sequencing were performed by Eurofins Genomics. All DNA plasmids were purified via miniprep (Omega Bio-Tek) and sequenced to verify correct assembly. A Plasmid Editor (ApE, version 3.1.3) and SnapGene Viewer software (version 5.0.7) were used to facilitate primer design and sequence alignments. Polymerase chain reactions

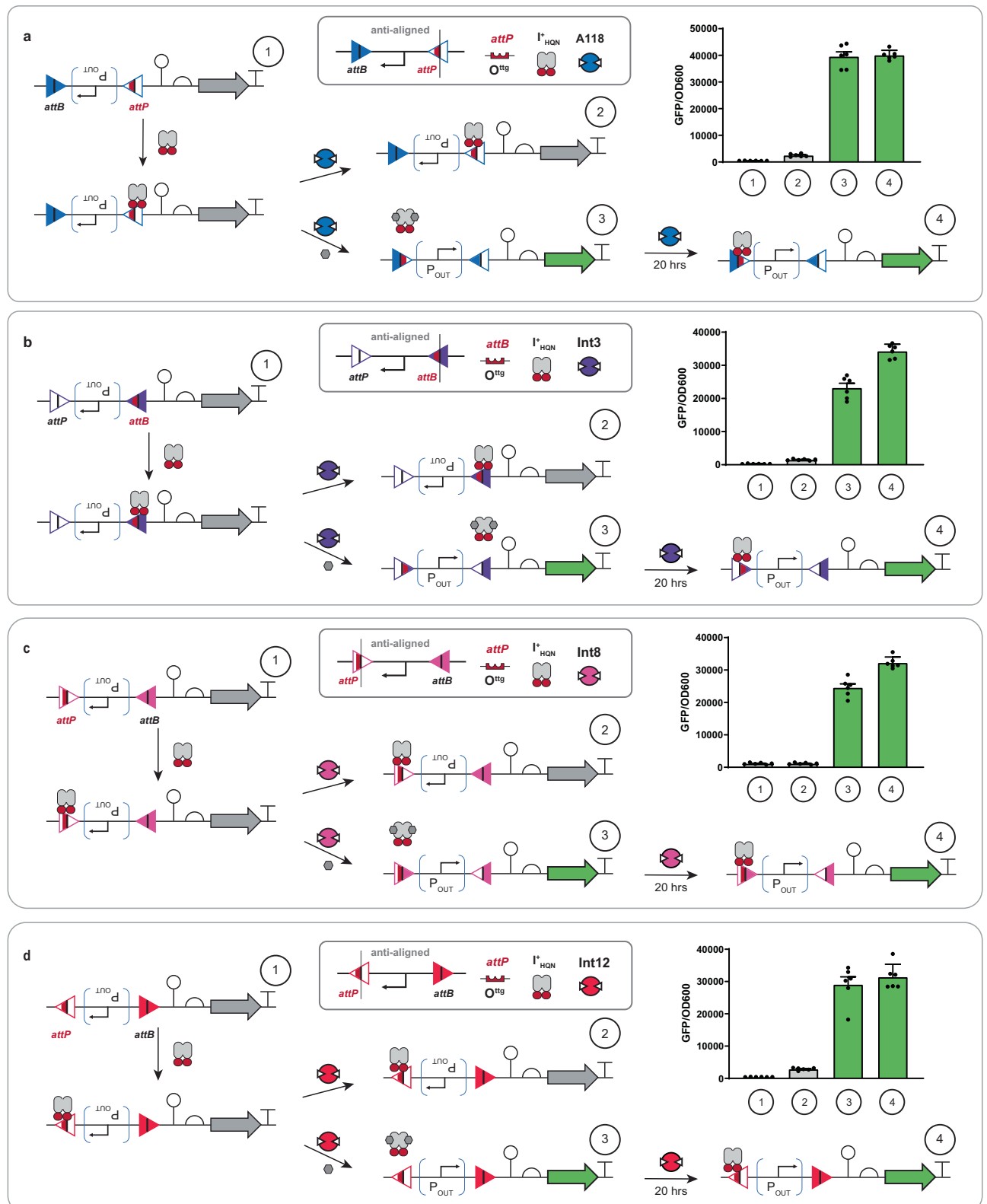

**Fig. 7 | Interception synthetic memory facilitating controlled DNA inversions.** Promoter inversions controlled by interception demonstrated for four different recombinases, **a** A118, **b** Int3, **c** Int8, and **d** Int12. For each subfigure (**a**–**d**), diagram (1) at top left shows a schematic of the corresponding promoter inversion circuit. In this state, the promoter is in the opposite orientation to transcribe GFP, corresponding to reduced fluorescence in the fluorescence assay data (shown in the right corner). As shown in diagram (2), adding $I^{+}_{HQN}$ repressor protects the circuit from recombinase mediated inversion via binding to the $O^{ttg}$ operator at position P + 1. As shown in diagram (3), when IPTG is added to induce $I^{+}_{HQN}$ to unbind from the operator, the recombinase (variable) can bind and recombine the att sites, inverting the promoter resulting in the expression of GFP. To demonstrate that repressor binding to the inverted circuit does not impede GFP expression once the inducer is absent, cells in state (3) were diluted 1:200 in fresh minimal media and grown for 20 additional hours, then assayed again at state (4). General anti-alignment configurations are given in Supplementary Fig. 1. Source data are provided as a Source Data file. Data in (**a**–**d**) represent the average of *n* = 6 biological replicates. Error bars correspond to the SEM of these measurements.

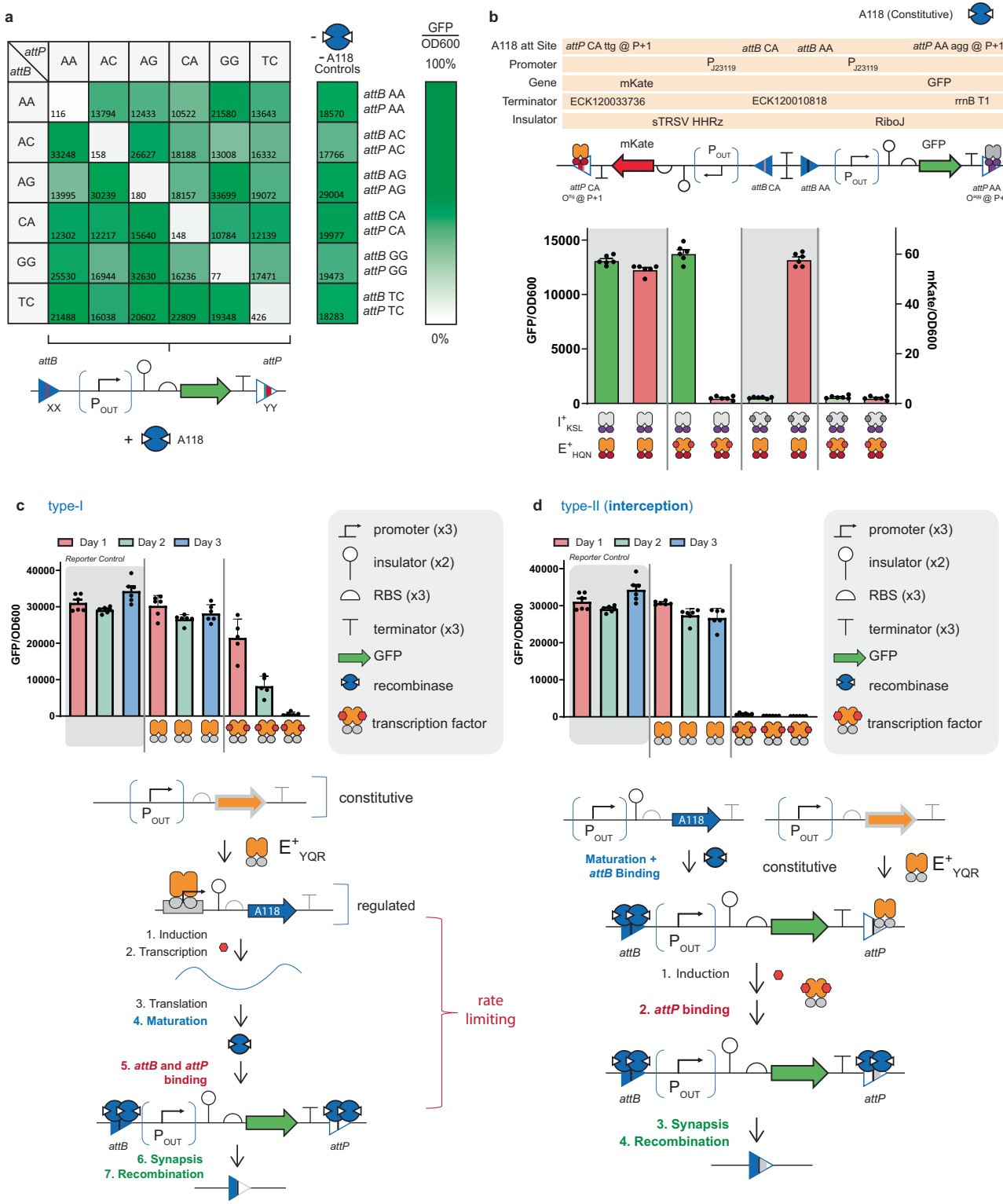

(PCR) were performed using Phusion High-Fidelity PCR Master Mix with HF Buffer or GC Buffer (NEB), or using Q5 Polymerase Master Mix (NEB) on a C1000 Touch Thermal Cycler (Bio-Rad). BsaI-HF®v2 restriction enzyme (NEB) was used to construct operator position libraries, and BfuAI restriction enzyme (NEB) was used to construct three-TF plasmids as described below. AvrII restriction enzyme (NEB), AatII restriction enzyme (NEB), NdeI restriction enzyme (NEB), and PacI restriction enzyme (NEB) were used to tune the expression of E⁺ and I⁺ on the two-TF plasmids used for nested transcriptional logic circuits as described below.

**Reporter plasmid constructs**

The reporter plasmids containing the deletion circuit architecture (shown in Fig. 2a) were constructed starting with the pZS*22-sfGFP plasmid reported in Richards et al.[33], featuring a low-copy-number pSC101* origin of replication and kanamycin resistance. The J23119 promoter and RiboJ segments were ordered as gene fragments from Eurofins Genomics and used to replace the LacIQ promoter on pZS*22 via Gibson assembly (NEB HIFI enzymes). Similarly, the two attachment sites (attB and attP) complementary to each recombinase tested were ordered as gene fragments from Eurofins Genomics and

**Fig. 8 | Engineering interception *att* orthogonality and memory kinetics.**
**a** Recombination matrix for A118 attachment-site pairs with matched (along the diagonal) and mismatched (off-diagonal) central dinucleotides. Rows correspond to *attB* sites having the central dinucleotide listed at left, and columns correspond to *attP* sites having the central dinucleotide listed across the top. Each box displays GFP output for a deletion circuit having the corresponding attachment sites in the presence of constitutive A118 expression as shown at bottom. The relative expression level of GFP is shown inside each box. Controls for attachment site pairs with matched central dinucleotides and no A118 expression are shown to the right of the matrix. Scale bar reference for GFP output is scaled to each row's maximum GFP value. **b** At left is shown a genetic schematic for a two-channel deletion circuit containing two fluorescent outputs (mKate and GFP). Below assay data is shown for this two-output circuit co-transformed with a constitutive I$^+_{KSL}$ and E$^+_{HQN}$ expression plasmid and a constitutive A118-expression plasmid under the four different INPUT conditions shown at bottom (see "Methods", "Two-output circuit assay" for details). **c** Kinetic assay data over 3 days is shown for the type-I memory circuit shown in Fig. 1f. On the plot at left, boxed in gray, is control data for cells containing only the GFP reporter plasmid (Reporter Control). The center three bars represent the circuit with no inducer (cellobiose) added and correspond to A118 transcription being repressed over 3 days. The three bars at right represent the circuit with inducer added and correspond to A118 transcription being on for 3 days. **d** At right, kinetic assay data over 3 days is shown for type-II (interception) memory shown below, with the same inducer conditions described in (**c**) (see "Methods", "Recombinase 3-day kinetic assays" for more information). Note: The icon for the recombinase is given as a monomer. Source data are provided as a Source Data file. Data in (**a**–**d**) represent the average of *n* = 6 biological replicates. Error bars correspond to the SEM of these measurements.

added upstream and downstream of sfGFP via Gibson assembly (NEB HIFI enzymes). As described in Fig. 2j, minimal 16 bp LacI-family operators were added in the place of attachment site DNA via site-directed mutagenesis using Phusion DNA polymerase with HF buffer (NEB), facilitated by the NEBuilder software (version 2.8.2, https://nebuilder.neb.com/#!/) followed by treatment with KLD Enzyme Mix (NEB). Attachment half-site deletions were performed via PCR followed by treatment with KLD enzyme mix. For half-sites "B2" and "P1" (see Fig. 2), random DNA spacers were added in place of the B2 and P1 attachment half-sites; these were designed using UCR's Random DNA Sequence Generator Tool (http://faculty.ucr.edu/~mmaduro/random.htm). To construct the "nested-logic" reporter construct (shown in Supplementary Figs. 20–22), the reporter plasmid containing the deletion circuit architecture for the A118 recombinase described in "Recombinase-expression constructs" was linearized to delete the J23119 promoter. Alternative promoter regions based on the pTrc promoter but modified to include the O$^{gtg}$ YQR operator in either the "core" or "proximal" position were amplified from plasmids available in lab, developed for Rondon et al.[3] These alternative promoters were added to the reporter plasmid via Gibson assembly (NEB HIFI) and sequence verified.

### Recombinase-expression constructs
Recombinase-expression plasmids were constructed using a pLacI (Novagen) backbone featuring a medium-copy-number p15A origin and chloramphenicol resistance. A118, Bxb1, and TP901 recombinases were independently amplified for Gibson assembly (NEB HIFI) from the pNR220 plasmid (a gift from the Lu lab at MIT), int2, int5, and int8 were amplified for Gibson assembly from the pCis_2 + 7 + 8 + 5 plasmid (Addgene 60588), int3 was amplified from pIntegrase_3 (Addgene 60575) and int12 was amplified from pIntegrase_12 (Addgene 60583)[12]. The expression level of recombinases was tuned using RBS tuning and promoter tuning. RBS tuning was facilitated by the Salis lab's RBS Library Calculator (version 2.0, https://salislab.net/software/)[37,38], and performed using site-directed mutagenesis with variable-codon primer tails (Eurofins Genomics). Variable-strength promoters were taken from Wang et.al., ordered as genestrands from Eurofins genomics, and added via Gibson assembly.

### Transcription-factor-expression constructs
Transcription-factor-expression plasmids were constructed using a pTB146 backbone (a gift from the Xiong lab at Yale University) containing a medium-high copy number ColE1 origin and M13 origin (with copy number suppressed by the Rop protein) and ampicillin resistance. Chimeric transcription factors were sourced in-house from plasmids developed by Rondon et al.[3,35] and Groseclose et al.[2,34] and added downstream of a constitutive LacI promoter via Gibson assembly (NEB HIFI). F$^+$ and F$^{A(1)}$ were placed downstream of the constitutive LacIQ promoter to increase the transcription rate about 10 fold. Mutations to the DNA-binding domains were introduced as needed via site-directed mutagenesis using Phusion DNA polymerase with HF buffer (NEB) targeting nucleotides outlined in Rondon et al. Plasmids containing two TFs were constructed via Gibson assembly (NEB HIFI). Plasmids containing three TFs were constructed by amplifying individual TFs using primers containing BfuAI recognition sites (NEB) from single-TF plasmids, subcloning these in a pUC19 cloning vector (NEB N3041S) via Gibson assembly (NEB HIFI), and assembling the three parts into one destination vector on the pTB146 backbone through Golden Gate Cloning.

### Microwell plate assay
The protocol for the fluorescence-based microwell plate assay was taken from Richards et al.[33] Each relevant plasmid was chemically transformed into 3.32 *E. coli* cells. Transformants were selected on LB Miller agar (Fisher Scientific) plates containing the corresponding antibiotics (and inducers at 1 mM or 10 mM, for anti-repressor experiments), then six replicates were grown for 8 h in a 96-well clear, flat-bottomed assay plate (Costar) sealed with a Breathe-Easier membrane (Midwest Scientific), shaking at 300 RPM at 37 °C in LB Miller in a Fisher shaker (Fisher Scientific MaxQ400) with appropriate antibiotics (and relevant inducers, for anti-repressor experiments). One µl from each of the six culture replicates was then diluted in 200 µl of supplemented 1X M9 Minimal Media (containing the relevant set of antibiotics and inducers) in a 96-well clear, flat-bottomed culture plate (Costar). Plates were sealed with Breathe-Easyy membranes (Midwest Scientific) to prevent evaporation and grown for 16 h, shaking at 300 RPM at 37 °C in a Fisher shaker (Fisher Scientific MaxQ400). 150 µl of cells from each well were transferred to a 96-well black-sided, clear-bottomed assay plate (Costar). Fluorescence and optical density (OD$_{600}$) was measured via plate reader (Molecular Devices SpectraMax M2e) using an excitation wavelength of 485 nm and an emission wavelength of 510 nm (for sfGFP reporters). Data was collected with SoftMax Pro Software (version 7.0.3, Molecular Devices). Wells containing M9 Minimal Media and relevant antibiotics and inducers with no cell inoculations were used as blanks; the average fluorescence of the six blanks for each condition was subtracted from each sample fluorescence intensity reading. Similarly, the average optical density of the six blanks for each condition was subtracted from each sample optical density reading. Blank-compensated fluorescence data was then normalized to blank-compensated optical density data for each sample, with an aim to quantify average fluorescence per cell. Data analysis was performed in Microsoft Excel (2021) and GraphPad Prism (version 9.3.1).

### Two-output circuit assay
The mKate/GFP two-output circuit shown in Fig. 8b was assayed following an additional 24-h media passage with no inducers to demonstrate maintenance of memory state and allow the degradation of excess fluorescent proteins from cells following fluorescent protein circuit deletion. Briefly, variants were subjected to the same growth

conditions given in the Microwell Plate Assay section. After the 16 h growth step in minimal media with and without the relevant inducers, replicates were diluted 1:200 in LB Miller with appropriate antibiotics and no inducers and shaken for an additional 8 h in a 96-well clear, flat-bottomed assay plate (Costar) sealed with a Breathe-Easier membrane (Midwest Scientific), shaking at 300 RPM at 37 °C. One μl from each of the six culture replicates was then diluted in 200 μl of supplemented 1X M9 Minimal Media (containing the relevant set of antibiotics and no inducers) in a 96-well clear, flat-bottomed culture plate (Costar). Plates were sealed with Breathe-Easy membranes (Midwest Scientific) to prevent evaporation and grown for 16 h, shaking at 300 RPM at 37 °C in a Fisher shaker (Fisher Scientific MaxQ400). 150 μl of cells from each well were transferred to a 96-well black-sided, clear-bottomed assay plate (Costar). Fluorescence and optical density (OD600) were measured via plate reader (Molecular Devices SpectraMax M2e) using an excitation wavelength of 485 nm and an emission wavelength of 510 nm (for sfGFP) and an excitation wavelength of 588 nm and an emission wavelength of 635 nm (for mKate).

### Recombinase 3-day kinetic assays

For testing the kinetics of transcriptionally-regulated A118 expression, 3.32 *E. coli* cells were transformed with the pSK012 transcriptionally regulated A118 plasmid along with the relevant TF-expression plasmids and the pSK153 reporter plasmid (having the $O^{agg}$ operator, which is orthogonal to the TF's YQR DNA-binding domains, at the P + 1 position). For testing the kinetics of interception-regulated A118 function, 3.32 *E. coli* cells were transformed with the pSK001 constitutive A118 expression plasmid along with the relevant TF-expression plasmids and the pSK148 plasmid (having the $O^{gtg}$ operator, which the TF's YQR DNA-binding domains bind to for interception, at the P + 1 position). In both cases, data collection was performed at three timepoints labeled "Day 1", "Day 2", and "Day 3." For the Day 1 timepoint, cells were precultured and assayed as described in "Microwell plate assay". For the Day 2 and Day 3 timepoints, the same preculture and assay conditions were applied, except the cells were transferred to preculture (being diluted 1:200) in LB Miller from the previous assay plate rather than from the initial petri dishes.

### Recombinase plate reader kinetic assays

For testing the kinetics of transcriptionally-regulated A118 expression, 3.32 *E. coli* cells were transformed with the pSK012 transcriptionally regulated A118 plasmid along with the relevant TF-expression plasmids and the pSK153 reporter plasmid (having the $O^{agg}$ operator, which is orthogonal to the TF's YQR DNA-binding domains, at the P + 1 position). For testing the kinetics of interception-regulated A118 function, 3.32 *E. coli* cells were transformed with the pSK001 constitutive A118 expression plasmid along with the relevant TF-expression plasmids and the pSK148 plasmid (having the $O^{gtg}$ operator, which the TF's YQR DNA-binding domains bind to for interception, at the P + 1 position). In both cases, transformants were precultured and diluted in minimal media conditions as outlined in the "Microwell plate assay" section. Assay plates were then placed in a plate reader (Molecular Devices SpectraMax M2e) and subjected to the following kinetic assay protocol: (1) set and hold temperature at 37 °C, (2) shake for 4 min, (3) read optical density (OD600), (4) read fluorescence using an excitation wavelength of 485 nm and an emission wavelength of 510 nm, (5) wait 3 min, (6) repeat (for a total of 1000 cycles).

### Recombinase RBS library construction

Recombinase RBS libraries designed in Salis lab's RBS Library Calculator were chemically transformed in NEB DH5-α Chemically Competent *Escherichia coli*, and scraped from plates and collectively miniprepped, yielding one DNA solution of RBS library. Then recombinase RBS library DNA was transformed in *E. coli* strain 3.32 with reporter plasmids and transcription factor plasmids. To cover the

variants, 96 transformants were picked and inoculated in a 96-well preculture plate (Costar) containing LB Miller with appropriate antibiotics, and shaken at 300 RPM at 37 °C in a Fisher shaker (Fisher Scientific MaxQ400) for 8 h. A 96-pin replicator (Boekel Sci) was used to inoculate these samples in two 96-well black-sided, clear-bottomed assay plates (Costar), each containing 1X M9 Minimal Media and antibiotics, and one containing the appropriate ligand. Assay plates were sealed with Breathe-Easy membranes (Midwest Scientific) to prevent evaporation and grown for 20 h, shaken at 300 RPM at 37 °C. To characterize the performance of the recombinases, the assay plates with and without ligand were compared and analyzed. Normalized fluorescence data as described in "Microwell plate assay" was used to analyze performance. Recombinase plasmids were then extracted by PCR (Phusion High-Fidelity PCR Master Mix with HF Buffer (NEB)), followed by treatment with KLD Enzyme Mix (NEB). Extracted RBS variants were then sequenced and analyzed through Salis lab's RBS Library Calculator. Finally, recombinases with tuned RBS were assayed again in sixplicate to confirm the tuned recombinase performances.

### Library construction for testing operator position within *attP* sites

Operator libraries were designed in Excel to include a unique 4 bp barcode downstream of the attachment site and ordered as oligopools from IDT Oligo Analyzer (version 3.1, https://www.idtdna.com/calc/analyzer). Oligopools were amplified with primers containing BsaI recognition sites to enable Golden Gate library assembly; BsaI-HF®v2 restriction enzyme (NEB) was used. The set of plasmids thus constructed was transformed into NEB 5-α Chemically Competent cells and selected on LB agar supplemented with kanamycin. Transformants were counted to quantify library coverage using equation 2 from Patrick et al.[39]; all libraries used had expected coverage >95%. Transformants were then scraped from plates and collectively miniprepped, yielding one DNA solution containing the library. The cells were grown as described in "Recombinase RBS library construction." Easy-Fluorescence measurements were recorded as described in the "Microwell plate assay" section. Samples exhibiting target phenotypes were individually grown in LB Miller and sequenced; the barcode was used to identify operator position. Positions thus identified were assayed in sixplicate as described in the "Microwell plate assay" section.

### PCR and gel-electrophoretic genotyping

Colony PCR was conducted to confirm the genetic deletion of recombinase-mediated deletion circuits. Cells containing deletion circuits for recombinases A118, Bxb1, Int2, Int3, Int5, Int8, Int12, and TP901 (shown in Supplementary Fig. 4) transformed both with and without the corresponding recombinase-expression plasmids were precultured and assayed as described in the "Microwell plate assay" subsection. Colony PCR was performed directly from the assay plates following fluorescence reading; 1 μl of cells diluted in 40 μl of DI H2O was used as template for these reactions. Primers were designed to bind upstream and downstream of the deletion region such that an undeleted circuit would generate a PCR product of ~1300 bp, and a deleted circuit would generate a PCR product of ~200 bp (with slightly different spacing depending on the length of each recombinase's attachment sites). PCRs were carried out with a C1000 Touch Thermal Cycler (Bio-Rad) and preceded by a 10-min time period at 98 °C to lyse cells. The PCR products were analyzed by electrophoresis on a 2.0% agarose gel. The sequences of the colony PCR primers are as follows: GenotypeF 62: CATCCAGTTTACTTTGCAGGG, GenotypeR 62: GATAA CAAACTAGCAACACCAGAAC.

### Flow cytometry

Fluorescence analysis was performed with a Beckman Coulter Cytoflex S flow cytometer. Cells were grown as described in "Microwell plate

assay", then passaged for an additional 24 h with no inducer as described in "Recombinase 3-day kinetic assays." Cells were then diluted 1:19 into PBS with 2 mg/ml kanamycin and incubated for at least 1 h at room temperature to inhibit further protein synthesis. Cells were analyzed by flow cytometry software, CytExpert 2.5. Cells were processed at 10–30 ul/min and monitored through the FITC channel for GFP expression and (where applicable) the ECD channel for mKate expression. Events were gated by forward scatter area vs. side scatter area to eliminate debris and then gated by side scatter height vs. side scatter area to discriminate doublets. More than 10,000 events were collected for final analysis. The cytometry gain and the gating procedure is shown in Supplementary Fig. 27.

## Statistics and reproducibility
The sample size for each experiment is indicated in the figure legend, where appropriate. In general, experiment in cell culture were confirmed in six independent experiments. The sample size of $n = 6$ biological replicates (individual colonies) were performed per day, with select assays repeated on a second and third day to provide kinetic data. Biological replicates were randomized by picking single colonies from transformed chassis cells. Each colony represents a single biological replicate. Results are shown as the mean ± standard error of the mean (SEM). Statistical comparison between any two groups was achieved via a two-sided unpaired Student's t-test. Statistical analysis and data visualization were performed using GraphPad Prism (version 9.3.1) and Microsoft Excel (2021).

## Reporting summary
Further information on research design is available in the Nature Portfolio Reporting Summary linked to this article.

# Data availability
The authors declare that all data supporting the findings of this study are available within the paper and its Supplementary Information. The analyzed data and source data are available in Supplementary Data Files and Source Data. The sequences of the following plasmids are provided in GenBank and as Source Data with respective accession numbers: pSK001–pSK012 (OR187764–OR187775), pSK101–pSK173 (OR187776–OR187811), pSK201–pSK275 (OR187812–OR187829) https://www.ncbi.nlm.nih.gov/nuccore/OR187764-OR187829. Source data are provided with this paper.

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

## Acknowledgements

This work was supported by National Science Foundation grants BIO 2319231, MCB 2123855; GCR/CBET 1934836; MCB 1921061; CBET 1844289; CBET 1804639 and MCB 1747439 all awarded to C.J.W.

## Author contributions

A.E.S., D.K., and C.J.W. conceived the study and designed the experiments; A.E.S., D.K., and P.T.M. performed experiments; A.E.S., D.K., P.T.M., and C.J.W. analyzed the data; A.E.S., D.K., and C.J.W. wrote the paper with input from all the authors.

## Competing interests

The authors declare no competing interests.
