## [Peer Review File · Nature Communications]

Reviewers' Comments:

Reviewer #1:

Remarks to the Author:

The manuscript by Short et al. presents a novel strategy for regulation of synthetic genetic memory operations (i.e. site-specific recombinase-promoted DNA rearrangements). Briefly, deletion or inversion of a DNA segment by recombination between two flanking sites can be blocked ('intercepted') by binding of an engineered transcription factor (TF) to an operator sequence embedded within one of the sites, preventing access of the recombinase. The work thus provides a new set of systems which, when combined with the Wilson group's earlier (published) development of an impressive range of systematically engineered TF repressors and antirepressors, should substantially expand the parts and strategies available for creation of 'intelligent' genetic circuits.

The experimental work appears to be of a very high standard. The experimental systems have been carefully designed, with proper consideration given to possible confounding factors, the methods are adequately described, the results are presented clearly, and analysis of the results (including statistics) all seems to be appropriate. I have no doubts that the data are sound and their interpretation/conclusions justified.

Having said all these positive things, my main criticism of the manuscript is in the presentation of the main text. Although it mostly in good error-free English, it is very difficult to make sense of in many places; many readers are likely to quickly get lost and confused. Two major problems are that the Introduction does not adequately describe the principle of the systems the authors aim to construct (including their previously developed synthetic TFs), and there is no proper Discussion at the end; although there is a 'Discussion' heading, the text just seems to go into more results. There is also some quite odd/idiosyncratic use of language throughout, which makes it harder to understand. Specifically, I would highly recommend:

(a) that the authors begin the Introduction by explaining in easy-to-understand terms how they propose to create a new type of regulatable genetic memory system. The explanation should be accompanied by a clear and simple Figure, including a diagram showing how a recombinase dimer binds to the (att) site, and how the aim is to create a site that still binds a dimer but one half-site contains an operator sequence such that TF binding will block binding of one recombinase subunit, abolishing recombination activity.

(b) Inclusion of a small figure or Table illustrating the 'design principles' of the authors' set of TF repressors and antirepressors, plus their operators; i.e. a 'lookup table' that readers could refer to so they could understand the results figures better. See also comments below on TF and operator names.

(c) A proper (even if only quite brief) Discussion section summarizing what the results have shown and their implications for development of synthetic genetic memory systems.

Comments

(1) Introduction first paragraph. The authors' definition of recombinases ('briefly, recombinases are...') is inaccurate and could cause confusion. All the site-specific recombinases used here are serine integrases, a specific subclass of recombinases; other subclasses don't conform to this definition. Also, it would be best to include the term 'site-specific' at least once (e.g. in this paragraph), as the term 'recombinase' is often used for other unrelated types of enzyme.

(2) Introduction second paragraph, and throughout the manuscript. I think the term 'intercalating' (or 'intercalated', etc.) isn't very appropriate for what the authors are doing to the att sites. It implies that the operator sequence is inserted into the att site sequence, creating a longer site, whereas the operator sequence is actually substituting for a sequence of the same length in the att site. I think 'substituting for' or 'replacing' att site sequence would be clearer.

(3) (Minor point) Introduction second paragraph, and throughout the manuscript. 'intimate' (and

derivatives) is used in a quite strange/unconventional sense. I would suggest (e.g.) 'propose'. Also, 'mitigate' is used oddly in a few places later in the ms.

(4) The whole Introduction. As noted above, it's not at all reader-friendly. I think it could be much improved, especially with some text and a figure explaining the design and basis of the authors' new memory systems.

(5) Results, first paragraph. The authors have done some nice experiments to identify att sites that are still active when a half-site is deleted. From previous work it's clear that serine integrases must form a tetramer to catalyze recombination, so these experiments (and the ones that follow where an operator sequence replaces part of a half-site) presumably identify systems that can retain the 'fourth subunit', either because of relaxed DNA binding specificity or strong integrase-integrase interactions. The DNA sequence effects will depend on which basepairs (and how many) of the original att site remain the same in the modified sites; the identity of some basepairs might be much more important than others. Some readers might be interested in this aspect of the work, so I would suggest that the authors could include simple alignment figures in the Supplementary material showing the sequences of the original att sites and all the variants, highlighting the basepairs that are different.

(6) Results second paragraph. From here onwards the manuscript refers extensively to the engineered TFs (such as E+HQN etc.) that the Wilson group reported in previous publications (ms. refs 2 and 3). But they are not introduced or explained to the reader. I think the authors should not assume that the readers will understand what these TFs are, and should give a clear summary of them (maybe with a 'lookup table' figure; see above). Furthermore, the naming of these TFs and their cognate operator sequences is likely to be very confusing. The big letters (E, R, C etc.) and the accompanying symbol (e.g. +) are OK - they have to be called something. The 'specificity' triplets of letters are more confusing, as they refer to the DBD mutations from wtLacI, and don't correlate in any obvious way with the operator sequence names (Ogac, OSYM, etc.) which are related to the mutations of the canonical operator. The colours in the Figures help a little, but it's still very hard to follow. I'm sure that the authors could think of a way to make this easier for the readers, for example by giving related names to the DBDs and their cognate operator targets - but at the least, an explanatory diagram/table would help.

(7) The Results section on 'Engineering permissive memory circuits' is particularly hard to read. The authors should try to ensure that a non-expert reader can understand it.

(8) The experiments described in the Section 'Interception with nested decision-making' (Figure 6 data) don't seem to add much to the manuscript - it seems quite obvious (from the previous data shown) that a setup like this could be constructed. I would suggest deleting it, or transferring it to Supplementary - especially if this would increase space in the main text for clearer descriptions of the work as outlined above.

(9) Discussion. As noted above, the text under this heading doesn't read like a Discussion at all; it's more results. I think it is important to have at least a brief Discussion/Conclusions section summing up what the experiments have shown, and outlining how the authors' novel memory systems could be implemented for practical applications. The current 'Discussion' material should really be in the Results section.

(10) (minor point) I am guessing that the symbol used for the recombinase has two little arrows on it to indicate a recombinase dimer. That's OK, but see point (15) below.

(11) (minor point) Figure 2a. The symbol shown for the operator is not actually used in part a (though it is used in b-e).

(12) The diagrams in Figure 5a showing both repressors binding to the operator site in the right-hand

atts is potentially misleading; presumably only one repressor binds at any one time. It might be enough to clarify this in the Figure legend.

(13) Figure 7. It would be helpful to label the parts (a-d) with the relevant recombinase (A118, int3, etc.). Also, is something different about the construct used in part b (which is labelled attB OP in the box at the top) from the others that are labelled attP OP? Perhaps I don't understand the symbols/colour scheme; some more explanation in the legend would be good. As a general point for all the Figures, it might clarify things for the reader if the att sites were labelled P and B on the actual diagrams (a small letter below each site would suffice). Also, the product site could be labelled L (or R).

(14) Figure 8 part a. I think quantitative data should be given for the experiments with altered central 2 bp sequences, not just colours. The diagrams on the right side of part (a) seem repetitive/unnecessary.

(15) Figure 8 part c. In this diagram the recombinase symbol is being used to represent a monomer (see point (10) above).

(16) Figure 8 parts c and d. The layouts of these parts could be made more comparable to each other, so it's easier for the reader to see what is different. For example, part d could show the substrate with recombinase bound to both sites, prior to synapsis and recombination.

(17) Figure 8 part e. The barcharts seem to suggest that recombination following "Type I" induction is very slow (taking days for GFP signal to decline to near zero). Is there an explanation for this?

(18) Supplementary Figure 1. The layout of the diagrams isn't very clear. The truth tables are arranged unconventionally (INPUT and OUTPUT horizontal rather than vertical), which is OK if it's labelled unambiguously; but in the bottom panels there's an 'OUTPUT' label, not in the top ones. Also it's not very obvious (despite the colouring red) that 'INPUT removed' refers just to the final column. Also, I think there may be an error in the legend to part b; should it read "...reverts to an OUTPUT with a value of 1..." (not "0")?

(19) Supplementary Figure 3. In the att site "omission" constructs, it would be good to show the sequences that replace the deleted regions (see point (5) above).

(20) Supplementary Figure 4. Just a minor criticism; PCR is not a very good way to get a quantitative measurement of the extent of deletion (for example, the short 'deletion' PCR product could be favoured). It would be quite straightforward to get a better estimate by making plasmid DNA from the cells and analysing by gel electrophoresis after a restriction digest.

(21) Is Supp. Table 1 in the right place? (i.e. between Supp. Figures 5 and 6)

(22) Supplementary Figure 6. In the lower part of the figure, why show data on both left and right barcharts that are exactly the same and are not involved in the 'RBS tuning'?

(23) Supplementary Figure 8. As point (22) above, except that for the 'red' data, the two bars look identical except for one data point that has moved much higher. Is this OK?

(24) Title of Supplementary information section says "...biotic memory..." whereas main text title is "...synthetic memory...".

Reviewer #2:

Remarks to the Author:

This paper describes an interception synthetic memory operator that pairs transcriptional programming with recombinase-based genetic circuits. To develop the system, they engineered the recombinase recognition site to contain the transcription operator such that the transcription factor binding can inhibit recombinase access to recognition site. Such design was then screened and verified with different recombinases and transcription regulating systems, both activators and inhibitors. They also develop combinational (2-input) genetic circuits and nested AND- and NOR- gate with the interception circuits. Moreover, through a two-channel deletion circuit, they proved that the engineered system could perform with higher speed and larger capacity than the previous generations of recombinase-based memory.

Combining transcription factors with recombinase to generate genetic logic gate has been used before, but the paper has the main strengths of being innovative in that instead of controlling transcription, they controlled recombinase activity post-translation with the transcription factors. Such design, as stated in the paper, allows a faster response to the induction. In addition, the paper also provides useful design rules based on analysis from extensive screening data for various recombinases, DNA operators, and regulatory domains. The paper probably has done a good job for the system design and characterization. I do believe that the paper can make an impact on synthetic biological system design for myriad applications. Therefore, I support publication in Nature Communications after addressing the comments below.

Major comments

Leaky recombination: Their experiment analyzes the percentage of un-induced population under with and without inducers (Supplementary Figure 5 – Part 7). It shows that for some transcription factor and operator pairs, about half of the population under no inducer condition was recombined, indicating there are leaky recombination events for those systems. It is uncertain how the percentage of the recombined population changes over time. Will the percentage of the recombined population gradually increase and thus damage the system's overall performance?

Versatility of the system: In addition to the transcription factors that were engineered by the researcher lab before, can the system be applied to other naturally existed transcription factors, such as tetR and smtR?

Ribozymes: In the caption for the schematic of reporter design in Figure 1, it is mentioned that there is a ribozyme after the promoter. The reason of why a ribozyme exists in the reporter plasmid was not explained.

Minor comments

Figure 1a and any other figure to show reporter plasmid design: besides the recombinase binding sites, please also label the other components on the plasmid, such as the promoter and RBS (although they are mentioned in the caption).

Figure 3: The transcription factor color scheme makes it difficult to differentiate between different transcription factors – probably increase the thickness of the outline might help, whose color is the indicator of various TFs.

Figure 4g and Figure 8e: for induced and noninduced conditions, if the difference between two bars is

not obvious, it is better to perform a statistical test.

Reviewer 1 Comments:

The manuscript by Short et al. presents a novel strategy for regulation of synthetic genetic memory operations (i.e. site-specific recombinase-promoted DNA rearrangements). Briefly, deletion or inversion of a DNA segment by recombination between two flanking sites can be blocked ('intercepted') by binding of an engineered transcription factor (TF) to an operator sequence embedded within one of the sites, preventing access of the recombinase. The work thus provides a new set of systems which, when combined with the Wilson group's earlier (published) development of an impressive range of systematically engineered TF repressors and antirepressors, should substantially expand the parts and strategies available for creation of 'intelligent' genetic circuits. The experimental work appears to be of a very high standard. The experimental systems have been carefully designed, with proper consideration given to possible confounding factors, the methods are adequately described, the results are presented clearly, and analysis of the results (including statistics) all seems to be appropriate. I have no doubts that the data are sound and their interpretation/conclusions justified. Having said all these positive things, my main criticism of the manuscript is in the presentation of the main text. Although it mostly in good error-free English, it is very difficult to make sense of in many places; many readers are likely to quickly get lost and confused. Two major problems are that the Introduction does not adequately describe the principle of the systems the authors aim to construct (including their previously developed synthetic TFs), and there is no proper Discussion at the end; although there is a 'Discussion' heading, the text just seems to go into more results. There is also some quite odd/idiosyncratic use of language throughout, which makes it harder to understand. Specifically, I would highly recommend: (a) that the authors begin the Introduction by explaining in easy-to-understand terms how they propose to create a new type of regulatable genetic memory system. The explanation should be accompanied by a clear and simple Figure, including a diagram showing how a recombinase dimer binds to the (att) site, and how the aim is to create a site that still binds a dimer but one half-site contains an operator sequence such that TF binding will block binding of one recombinase subunit, abolishing recombination activity. (b) Inclusion of a small figure or Table illustrating the 'design principles' of the authors' set of TF repressors and antirepressors, plus their operators; i.e. a 'lookup table' that readers could refer to so they could understand the results figures better. See also comments below on TF and operator names. (c) A proper (even if only quite brief) Discussion section summarizing what the results have shown and their implications for development of synthetic genetic memory systems.

Comment 1: Introduction first paragraph. The authors' definition of recombinases ('briefly, recombinases are...') is inaccurate and could cause confusion. All the site-specific recombinases used here are serine integrases, a specific subclass of recombinases; other subclasses don't conform to this definition. Also, it would be best to include the term 'site-specific' at least once (e.g. in this paragraph), as the term 'recombinase' is often used for other unrelated types of enzyme.

Response 1: Thank you for pointing this out. We have changed the text in the introduction: (i) specifying the use of serine integrases in this study, and (ii) using the term site-specific to further specify the recombinase classification.

Comment 2: Introduction second paragraph, and throughout the manuscript. I think the term 'intercalating' (or 'intercalated', etc.) isn't very appropriate for what the authors are doing to the att sites. It implies that the operator sequence is inserted into the att site sequence, creating a longer site, whereas the operator sequence is actually substituting for a sequence of the same length in the att site. I think 'substituting for' or 'replacing' att site sequence would be clearer.

Response 2: Done. We replaced 'intercalated' with the word substituted throughout the manuscript.

Comment 3: (Minor point) Introduction second paragraph, and throughout the manuscript. 'intimate' (and derivatives) is used in a quite strange/unconventional sense. I would suggest (e.g.) 'propose'. Also, 'mitigate' is used oddly in a few places later in the ms.

Response 3: Done. We have modified the manuscript accordingly.

Comment 4: The whole Introduction. As noted above, it's not at all reader-friendly. I think it could be much improved, especially with some text and a figure explaining the design and basis of the authors' new memory systems.

Response 4: Done. We have revised the introduction and prepared a new figure to make the introduction more reader friendly.

Comment 5: Results, first paragraph. The authors have done some nice experiments to identify att sites that are still active when a half-site is deleted. From previous work it's clear that serine integrases must form a tetramer to catalyze recombination, so these experiments (and the ones that follow where an operator sequence replaces part of a half-site) presumably identify systems that can retain the 'fourth subunit', either because of relaxed DNA binding specificity or strong integrase-integrase interactions. The DNA sequence effects will depend on which basepairs (and how many) of the original att site remain the same in the modified sites; the identity of some basepairs might be much more important than others. Some readers might be interested in this aspect of the work, so I would suggest that the authors could include simple alignment figures in the Supplementary material showing the sequences of the original att sites and all the variants, highlighting the basepairs that are different.

Response 5: Done. This is now part of Supplementary Fig. 2 and Supplementary Fig. 4. In addition, all sequences will be made available on GenBank.

Comment 6: Results second paragraph. From here onwards the manuscript refers extensively to the engineered TFs (such as E+HQN etc.) that the Wilson group reported in previous publications (ms. refs 2 and 3). But they are not introduced or explained to the reader. I think the authors should not assume that the readers will understand what these TFs are, and should give a clear summary of them (maybe with a 'lookup table' figure; see above). Furthermore, the naming of these TFs and their cognate operator sequences is likely to be very confusing. The big letters (E, R, C etc.) and the accompanying symbol (e.g. +) are OK - they have to be called something. The 'specificity' triplets of letters are more confusing, as they refer to the DBD mutations

from wtLacI, and don't correlate in any obvious way with the operator sequence names (Ogac, OSYM, etc.) which are related to the mutations of the canonical operator. The colours in the Figures help a little, but it's still very hard to follow. I'm sure that the authors could think of a way to make this easier for the readers, for example by giving related names to the DBDs and their cognate operator targets - but at the least, an explanatory diagram/table would help.

Response 6: Done, see Figure 1 and introduction. One thing to note we do not want to change the nomenclature as it has been defined elsewhere and we do not want to cause confusion with other published work. However, we have now provided an extensive explanatory diagram/table.

Comment 7: The Results section on 'Engineering permissive memory circuits' is particularly hard to read. The authors should try to ensure that a non-expert reader can understand it.

Response 7: Done. We have re-written this section to make it easier to read and interpret the results.

Comment 8: The experiments described in the Section 'Interception with nested decision-making' (Figure 6 data) don't seem to add much to the manuscript - it seems quite obvious (from the previous data shown) that a setup like this could be constructed. I would suggest deleting it, or transferring it to Supplementary - especially if this would increase space in the main text for clearer descriptions of the work as outlined above.

Response 8: Done. We have moved this section and figure to the supplement (see SI Fig. 9).

Comment 9: Discussion. As noted above, the text under this heading doesn't read like a Discussion at all; it's more results. I think it is important to have at least a brief Discussion/Conclusions section summing up what the experiments have shown, and outlining how the authors' novel memory systems could be implemented for practical applications. The current 'Discussion' material should really be in the Results section.

Response 8: Done. We have now made separate Results and Discussion sections as instructed. Note given the word number constraints (5,000) to achieve everything the reviewer requested we could only prepare a brief Discussion / Conclusion section.

Comment 10: (minor point) I am guessing that the symbol used for the recombinase has two little arrows on it to indicate a recombinase dimer. That's OK, but see point (15) below.

Response 10: Actually, each icon for the recombinase is a monomer – we have now indicated this in the figure legend.

Comment 11: (minor point) Figure 2a. The symbol shown for the operator is not actually used in part a (though it is used in b-e).

Response 11: In Figure 2a (now Figure 3a) the operator + attachment site iconography is defined in Figure 1j (now Figure 2j). To avoid redundancy we referenced 2j in 3a and elsewhere as needed.

Comment 12: The diagrams in Figure 5a showing both repressors binding to the operator site in the right-hand atts is potentially misleading; presumably only one repressor binds at any one time. It might be enough to clarify this in the Figure legend.

Response 12: The reviewer is correct. Figure 5a (now Figure 6a) has been corrected.

Comment 13: Figure 7. It would be helpful to label the parts (a-d) with the relevant recombinase (A118, int3, etc.). Also, is something different about the construct used in part b (which is labelled attB OP in the box at the top) from the others that are labelled attP OP? Perhaps I don't understand the symbols/colour scheme; some more explanation in the legend would be good. As a general point for all the Figures, it might clarify things for

the reader if the att sites were labelled P and B on the actual diagrams (a small letter below each site would suffice). Also, the product site could be labelled L (or R).

Response 13: Done. We have improved the labeling for Figure 7. However, we did not label the product site as we did not fully appreciate the reviewers instructions.

Comment 14: Figure 8 part a. I think quantitative data should be given for the experiments with altered central 2 bp sequences, not just colours. The diagrams on the right side of part (a) seem repetitive/unnecessary.

Response 14: Done, we have included the quantitative data in Figure 8a, and we have removed the right side of the figure.

Comment 15: Figure 8 part c. In this diagram the recombinase symbol is being used to represent a monomer (see point (10) above).

Response 15: YES – we have now indicated this in the figure legend.

Comment 16: Figure 8 parts c and d. The layouts of these parts could be made more comparable to each other, so it's easier for the reader to see what is different. For example, part d could show the substrate with recombinase bound to both sites, prior to synapsis and recombination.

Response 16: Noted – we have done our best to revise the figure to make type I and type more comparable. One thing to consider is that the circuit architectures are fundamentally different, accordingly an absolute comparison for each step will not be possible.

Comment 17: Figure 8 part e. The barcharts seem to suggest that recombination following “Type I” induction is very slow (taking days for GFP signal to decline to near zero). Is there an explanation for this?

Response 17: Done – also see “**Synthetic memory kinetics**” Results section. *“In all cases interception memory was significantly faster than the corresponding canonical (type-I) design – i.e., occurring nearly instantaneously opposed to hours or days. We attribute the increased rate of type-II memory over type-I memory to: (i) the maintenance of high levels of mature recombinase at steady-state, which (ii) leads to near instantaneous binding to the attB site. Moreover, the TF and recombinase are in a dynamic equilibrium at the substituted attP site. Once the TF (E^+_{YOR}) is induced, (iii) A118 recombinase binding and subsequent recombination occur near instantaneously.”*

Comment 18: Supplementary Figure 1. The layout of the diagrams isn't very clear. The truth tables are arranged unconventionally (INPUT and OUTPUT horizontal rather than vertical), which is OK if it's labelled unambiguously; but in the bottom panels there's an 'OUTPUT' label, not in the top ones. Also it's not very obvious (despite the colouring red) that 'INPUT removed' refers just to the final column. Also, I think there may be an error in the legend to part b; should it read “...reverts to an OUTPUT with a value of 1...” (not “0”)?

Response 18: We have revised the figure to improve interpretation – now see main text Figure 1a-b. Note, we have used the same truth table layout in previous publications; thus, to avoid any confusion we have decided to keep the horizontal layout. However, we have labeled the output as suggested and included a state diagram to improve interpretation.

Comment 19: Supplementary Figure 3. In the att site “omission” constructs, it would be good to show the sequences that replace the deleted regions (see point (5) above).

Response 19: Done. This is now part of Supplementary Fig. 2 and Supplementary Fig. 4. In addition, all sequences will be made available on GenBank.

Comment 20: Supplementary Figure 4. Just a minor criticism; PCR is not a very good way to get a quantitative measurement of the extent of deletion (for example, the short ‘deletion’ PCR product could be favoured). It would be quite straightforward to get a better estimate by making plasmid DNA from the cells and analysing by gel electrophoresis after a restriction digest.

Response 20: Excellent point. In fact, we did not intend to use this data as a quantitative measurement – rather a qualitative analysis. We have now made this clear in the manuscript. NOTE: we did conduct FACS as a quantitative measurement which is part of Supplementary Fig. 2.

Comment 21: Is Supp. Table 1 in the right place? (i.e. between Supp. Figures 5 and 6)

Response 21: We have moved SI Figure 1 to the end of the document to avoid breaking the continuity of the figure stream.

Comment 22: Supplementary Figure 6. In the lower part of the figure, why show data on both left and right barcharts that are exactly the same and are not involved in the ‘RBS tuning’?

Response 22: Done. We have removed all repeated data not involved in RBS tuning – now Supplementary Figure 5.

Comment 23: Supplementary Figure 8. As point (22) above, except that for the ‘red’ data, the two bars look identical except for one data point that has moved much higher. Is this OK?

Response 23: Done – corrected. Note: We have removed all repeated data not involved in RBS tuning (now Supplementary Figure 7). Accordingly, the data is no longer part of the figure.

Response 24: Title of Supplementary information section says “...biotic memory...” whereas main text title is “...synthetic memory...”.

Response 24: Done – we have corrected the title on the SI document to match the main text title.

Reviewer 2 Comments:

This paper describes an interception synthetic memory operator that pairs transcriptional programming with recombinase-based genetic circuits. To develop the system, they engineered the recombinase recognition site to contain the transcription operator such that the transcription factor binding can inhibit recombinase access to recognition site. Such design was then screened and verified with different recombinases and transcription regulating systems, both activators and inhibitors. They also develop combinational (2-input) genetic circuits and nested AND- and NOR- gate with the interception circuits. Moreover, through a two-channel deletion circuit, they proved that the engineered system could perform with higher speed and larger capacity than the previous generations of recombinase-based memory.

Combining transcription factors with recombinase to generate genetic logic gate has been used before, but the paper has the main strengths of being innovative in that instead of controlling transcription, they controlled recombinase activity post-translation with the transcription factors. Such design, as stated in the paper, allows a faster response to the induction. In addition, the paper also provides useful design rules based on analysis from extensive screening data for various recombinases, DNA operators, and regulatory domains. The paper probably has done a good job for the system design and characterization. I do believe that the paper can make an impact on synthetic biological system design for myriad applications. Therefore, I support publication in Nature Communications after addressing the comments below.

Comment 1: Leaky recombination: Their experiment analyzes the percentage of un-induced population under with and without inducers (Supplementary Figure 5 – Part 7). It shows that for some transcription factor and operator pairs, about half of the population under no inducer condition was recombined, indicating there are leaky recombination events for those systems. It is uncertain how the percentage of the recombined population changes over time. Will the percentage of the recombined population gradually increase and thus damage the system's overall performance?

Response 1: This is an excellent point. The short answer is the memory circuit improves overtime – that is the interception memory minus ligand / plus ligand increase substantially over a time course of three days. We demonstrate this in two separate examples SI Table 1 and SI Fig 15. Moreover, we show the interception memory circuits can support near perfect performance over a longer time course via FACS – see SI Figure 4 and main text Figure 8b (and related SI Figures 13-14).

Comment 2: Versatility of the system: In addition to the transcription factors that were engineered by the researcher lab before, can the system be applied to other naturally existed transcription factors, such as tetR and smtR?

Response 2: Yes. To illustrate this we have included an experiment demonstrating interception using aTc – see SI Figure 16.

Comment 3: Ribozymes: In the caption for the schematic of reporter design in Figure 1, it is mentioned that there is a ribozyme after the promoter. The reason of why a ribozyme exists in the reporter plasmid was not explained.

Response 3: Noted – we have now included a statement describing the purpose of the ribozyme in the figure legend.

Comment 4: Figure 1a and any other figure to show reporter plasmid design: besides the recombinase binding sites, please also label the other components on the plasmid, such as the promoter and RBS (although they are mentioned in the caption).

Response 4: Done. We have now provided a legend in Figure 1 describing all fundamental parts, in addition to other figures – provided we had sufficient space.

Comment 5: Figure 3: The transcription factor color scheme makes it difficult to differentiate between different transcription factors – probably increase the thickness of the outline might help, whose color is the indicator of various TFs.

Response 5: Noted. To help differentiate transcription factors we have included the nomenclature next to the icons whenever possible.

Comment 6: Figure 4g and Figure 8e: for induced and noninduced conditions, if the difference between two bars is not obvious, it is better to perform a statistical test.

Response 6: Noted. We have included the requested statistical analysis in the supplementary data section.

Comments to Editor:

In addition to the above we have reviewed and completed the relevant check lists, and have made all necessary changes to comply with *Nature Communications* publication standards - to the best of our knowledge. Below is a summary of completed tasks:

1. Editorial policy checklist (Returned)
2. Reporting requirements for life sciences research (Returned)
3. Data availability statements and data citations policy

Please let me know if you need any additional information.

Sincerely,

Corey J. Wilson, Ph.D.

Professor

AIMBE Fellow (Class of 2021)

Love Family Endowed Professor

NSF Growing Convergence Research Investigator

Georgia Institute of Technology

School of Chemical & Biomolecular Engineering